# Cyst-Wall-Protein-1 is fundamental for Golgi-like organelle neogenesis and cyst-wall biosynthesis in *Giardia lamblia*

Jacqueline A. Ebneter[1], Sally D. Heusser[1], Elisabeth M. Schraner[2], Adrian B. Hehl[1,*] & Carmen Faso[1,*]

The genome of the protozoan parasite *Giardia lamblia* is organized in two diploid nuclei, which has so far precluded complete analysis of gene function. Here we use a previously developed Cre/*loxP*-based knock-out and selection marker salvage strategy in the human-derived isolate *WB-C6* to eliminate all four copies of the Cyst-Wall-Protein-1 locus (*CWP1*). Because these loci are silenced in proliferating trophozoites and highly expressed only in encysting cells, CWP1 ablation allows functional characterization of a conditional phenotype in parasites induced to encyst. We show that encysting *Δcwp1* cells are unable to establish the stage-regulated trafficking machinery with Golgi-like encystation-specific vesicles required for cyst-wall formation but show morphological hallmarks of cyst development and karyokinesis. This 'pseudocyst' phenotype is rescued by transfection of *Δcwp1* cells with an episomally maintained CWP1 expression vector. Genome editing in genera *Giardia* and *Trypanosoma* are the only reported examples addressing questions on pathogen transmission within the Excavata supergroup.

[1] Institute of Parasitology, University of Zurich, Winterthurerstrasse 266a, CH-8057 Zurich, Switzerland. [2] Institute of Veterinary Anatomy, University of Zurich, Winterthurerstrasse 266b, CH-8057 Zurich, Switzerland. * These authors contributed equally to this work. Correspondence and requests for materials should be addressed to A.B.H. (email: adrian.hehl@uzh.ch) or to C.F. (email: carmen.faso@uzh.ch).

Deposition of an extracellular matrix is a common strategy for intestinal protozoan parasites to survive outside a host. The underlying matrix architecture of most of these environmentally resistant infectious forms consists of a tightly woven glycan mesh complexed with abundant and extensively cross-linked protein[1]. These mechanically and chemically resistant biopolymers protect parasites after excretion into the outside world and during stomach passage after ingestion by a new host. The cyst wall (CW) of *G. lamblia* is disassembled during stomach passage, allowing for emergence of a non-adherent precursor cell in the duodenum, which divides twice rapidly to form four flagellated trophozoites, which in turn initiate the new infection in the small intestine[2].

Encystation of *G. lamblia* trophozoites is induced by increased pH and lack of available lipids in the distal ileum. These conditions can be replicated *in vitro* where they initiate a differentiation process driven by the synthesis, trafficking, maturation and deposition of three cyst-wall proteins (CWPs 1–3), complexed with a unique β-1,3-linked N-Acetylgalactosamine (GalNAc) glycan polymer. Unlike the CW glycan synthesis pathway, which is up-regulated during encystation, the genes coding for CWPs are completely silenced in trophozoites and transcribed only in encysting cells[3–5]. Accumulation of CWPs 1–3 in unique Golgi-like organelles called encystation-specific vesicles (ESVs) after export via distinct endoplasmic reticulum (ER) exit sites initiates the stage-specific, *de novo* establishment of a regulated secretory pathway in encysting cells. After post-translational modifications of CWPs, and partitioning within ESVs, the mature cyst-wall material (CWM) is sorted into two biophysically and functionally distinct fractions that are secreted sequentially onto the plasma membrane to form a bi-layered CW[6]. Although the role played by the acidic tail of CWP2 in promoting condensed core formation in maturing ESVs has been described[6,7], we know very little about the individual contribution of each CWP to CW formation.

The most effective way for functional characterization of any given protein is to eliminate its corresponding gene and analyse the resulting phenotype. Classical insertion-based strategies combined with progeny segregation are effective in organisms amenable to Mendelian genetic analyses such as the model organisms *Saccharomyces cerevisiae* and *Arabidopsis thaliana*. Alternatively, and especially for polyploid organisms, the CRISPR-Cas9 system has emerged as a revolutionary tool for targeted genome editing[8–10]. However, protocols for its application to *Giardia* could not be established with the currently available tools. *Giardia*'s tetraploid status, the absence of a documented sexual cycle/gametes, and paucity of suitable selection markers present additional challenges and preclude a straight-forward application of classical insertion-based knock-out strategies.

In a previous report, proof-of-concept for the application of the Cre (Causes Recombination)/*loxP* (locus of crossing (x) over, P1) system to obtain selection marker-free, transgenic *G. lamblia* lines carrying insertions at defined genomic loci was presented[11]. However, we found that chromosomal insertion of a linearized construct by double cross-over occurs only at a single locus in a transfection experiment. Thus, disrupting four target gene alleles in two nuclei requires several sequential rounds of: (1) homologous recombination-mediated locus substitution with an antibiotic-resistance expression cassette, (2) selection of transgenic parasites and (3) excision of the resistance cassette using the Cre/*loxP* system. Here, we report on the impact of CWP1 ablation (open reading frame Gl50803_5638) on encystation and cyst formation after application of this sequential gene disruption strategy to target all four *CWP1* alleles. Induction of trophozoite encystation without a functional CWP1 (transgenic line *Δcwp1*) reveals fundamental defects in secretory organelle neogenesis, trafficking of the CWM and CW formation. These defects in encysting *Δcwp1* trophozoites and cysts can be fully complemented with a transfected wild-type *CWP1* locus. The early manifestation of encystation defects supports the model that CWPs act in concert to drive organelle neogenesis and maturation[12]. Interestingly, formation of wall-less *Δcwp1* 'pseudocysts' replicates a phenotype previously elicited by functional ablation of the GTPase Arf1 (ref. 13) and provides additional evidence for independent control of CWM synthesis and secretion, cell cycle exit, and morphological remodelling during *G. lamblia* stage-differentiation.

## Results

**Differentiating *Δcwp1* cells produce wall-less pseudocysts.** Genetic engineering of line *Δcwp1* (*Δcwp1-1::loxP, Δcwp1-2:: lox5171, Δcwp1-3::PAC, cwp1-4::NEO*) in a *WB-C6* background is based on sequential disruption of CWP1-encoding loci by recombination-driven insertion of antibiotic resistance cassettes. This was combined with removal of a puromycin-resistance expression cassette (*PAC*) flanked by lox sites *via* Cre-mediated excision analogous to a previously demonstrated proof-of-concept[11,14] (see also in the section 'Materials and Methods: Construction of CWP1KO1-4 plasmids and engineering of line *Δcwp1*' and Fig. 1a). A fluorescence microscopy-based phenotypic analysis of cysts depleted of 3 *CWP1* alleles out of 4 using monoclonal anti-CWP1 antibodies highlighted distinct defects in CWP1 deposition, distribution and CW integrity (Supplementary Fig. 1). Cysts depleted of either 1 or 2 alleles were phenotypically indistinguishable from wildtype cysts in terms of deposition of CWP1 on the cyst surface (Supplementary Fig. 1).

The elimination/disruption of all four *CWP1* loci was tested in each round by PCR analysis of the genomic (g) DNA, to confirm the designed genomic configuration of the final *Δcwp1* mutant line (Fig. 1a,b and Supplementary Fig. 2). Reverse transcription analysis of total RNA prepared from non-encysting (C), pre-encysting (PE; 44 h) and encysting (E; 8 h) cells, confirmed absence of CWP1-coding transcripts in the *Δcwp1* line (Fig. 1c). 3'RACE PCR amplification of a CWP2-coding fragment from the same cDNA as a control was positive (Fig. 1c). In line with the absence of intact *CWP1* loci and mRNA, immunoblotting-based detection of CWP1 using a specific commercial anti-CWP1 monoclonal antibody (mAb 5-3c) failed to develop any signal on extracts from 8-h-encysting *Δcwp1* cells, whereas extracts derived from 8-h-encysting WB cells showed a strong signal at the predicted size for monomeric CWP1 that is, *ca.* 25 kDa (Fig. 1d). A much weaker signal was detected in non-encysting WB trophozoites, in line with a small proportion of spontaneously encysting cells in non-induced populations grown *in vitro*[15–17]. As a phenotypic characterization of line *Δcwp1* we tested subcellular CWP1 deposition in an immunofluorescence assay (IFA). In contrast to cysts derived from wildtype *WB-C6*, *Δcwp1* cells induced to encyst *in vitro* failed to react with the same commercial anti-CWP1 mAb (Fig. 1e). This was consistent with the results of the gDNA, RNA and protein content analyses and confirmed complete ablation of CWP1 in *Δcwp1*. Surprisingly, cyst-like stages as defined by oval shaped cells lacking typical trophozoite cytoskeleton structures, that is, flagella and ventral disk, and having two nuclear pairs, a hallmark for cyst stages after secretion of the first CW layer, were still observed (Fig. 1e). However, these cells, hereafter referred to as pseudocysts, appeared non-refractile, translucent, and often damaged in bright field microscopy, suggesting complete absence of an extracellular matrix forming a CW. Taken together, these data

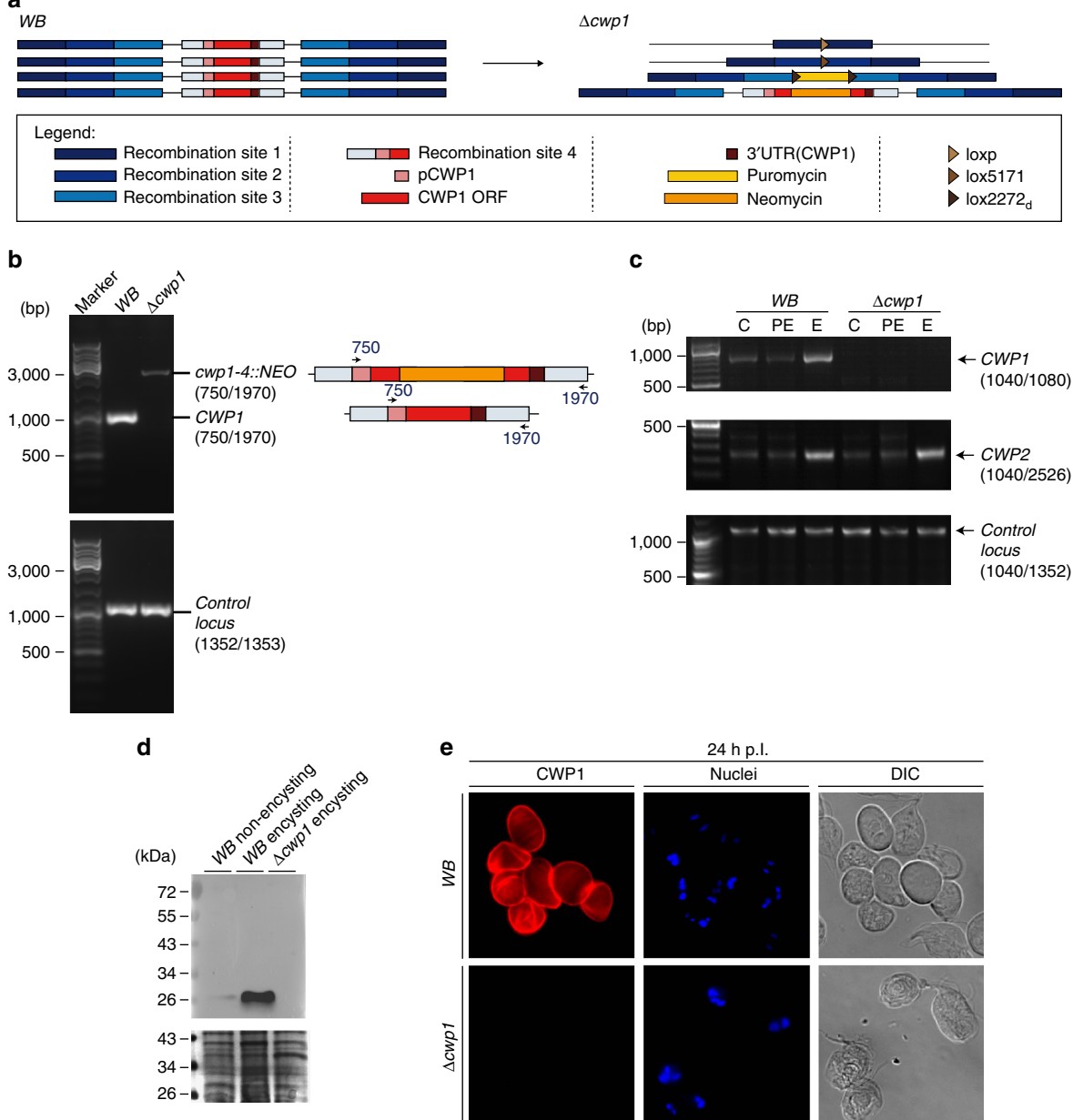

**Figure 1 | Generation and analysis of line Δcwp1.** (**a**) Configuration of line Δcwp1 compared to WB-C6. (**b**) PCR-based gDNA analysis demonstrating complete ablation of all CWP1 alleles in line Δcwp1 (upper panel). WB, wild type control sample. The higher molecular weight band found in lane Δcwp1 is the PCR product generated by the disruption of the allele in locus 4 with the neomycin resistance marker. gDNA control, PCR amplification of an unrelated gene locus (GGD gene model Gl50803_17161) (lower panel). Oligonucleotide primer numbers used for the analysis are indicated in parentheses (refer to Supplementary Table 1 for sequences). Marker lanes are labelled for main size bands (in bp). (**c**) RT-PCR and 3′RACE amplification of mRNA prepared from non-encysting (C), pre-encysting (PE) and encysting (E; 8 h p.l.) cells. A CWP1-specific product was amplified by RT-PCR in encysting WB cells (upper panel) but not in line Δcwp1. h p.l.: hours post induction of encystation. Controls: 3′RACE amplification of a CWP2-specific product from induced WB and Δcwp1 cells (middle panel); RT-PCR product (lower panel) from an unrelated mRNA (GGD gene model Gl50803_17161). Oligonucleotide primer numbers used for PCR amplification are indicated in parentheses (see Supplementary Table 1 for sequences). Marker lanes are labelled for main size bands (in bp). (**d**) Immunoblot analysis detects CWP1-specific signal in extracts prepared from 8-h-encysting WB cells but not from 8-h-encysting Δcwp1 cells. Loading controls are included beneath the blot. Marker lanes are labelled for main size bands (in kDa). (**e**) IFA highlights the absence of CWP1 (red) in cyst-like structures from line Δcwp1. Nuclei (blue) were stained with DAPI. DIC, differential interference contrast. Scale bar: 10 μm.

unequivocally confirm lack of CWP1 gene products in the Δcwp1 line and provide evidence for a decoupling of CWM secretion and morphological changes including loss of flagella and ventral attachment organelle. For further investigation we used these morphological features together with the presence of two nuclear pairs as the main diagnostic criteria for identification of pseudocysts in populations of differentiating Δcwp1 cells.

**Δcwp1 pseudocysts lose membrane integrity and viability.** Impermeability and resistance to water are hallmark characteristics of mature G. lamblia cysts and essential for transmission of Giardia to a new host. Water resistance has been primarily ascribed to the inner layer of the CW, composed of CWP3 and the small C-terminal processing product of CWP2, sealing off the polymerized outer wall from the inner side after secretion[6].

To characterize Δcwp1 pseudocysts in more detail we encysted WB-C6 and Δcwp1 cells using the 'high bile' protocol[18], which is optimal for producing viable cysts. Encysted cells were harvested and incubated in either cold PBS or distilled water for 18 h to assess osmotic resistance. WB-C6 cysts presented refractile CWs in bright field microscopy and aggregated readily, whereas Δcwp1 pseudocysts appeared shriveled, with no trace of a continuous refractile wall structure (Fig. 2a). Transmission electron microscopy (tEM) analysis of ultrathin sections of (pseudo)cyst-enriched samples from lines WB-C6 and Δcwp1 confirmed that Δcwp1 pseudocysts lacked the typical CW layer with fibrillar appearance of WB-C6 cysts (Fig. 2b). Given the lack of a wall structure and apparent loss of osmotic resistance in Δcwp1 pseudocysts, we used a cell viability staining assay based on acridine orange and ethidium bromide (AO/EB)[19] to test membrane integrity. Based on cytoplasmic labelling with acridine orange and exclusion of ethidium bromide, approximately 43% of WB-C6 cysts (n = 296) were scored as viable, whereas 98% (n = 162) of Δcwp1-derived cysts were non-viable (Fig. 2c). Taken together, this is direct evidence that CWP1 is necessary for formation of the CW structure, cyst viability and environmental resistance. However, the fate of the other CWPs in encysting Δcwp1 cells and pseudocysts remained unclear.

**CWP1 ablation abolishes ESV-dependent regulated secretion.** CWM trafficking, secretion and CW formation is detected with a commercial anti-CWP1 mAb, which could not be used to assess induction of the regulated secretory pathway in encysting Δcwp1 cells. Apart from the absence of the CW in fully formed pseudocysts, the most striking phenotype of encysting Δcwp1 cells was the lack of condensed ESVs in bright field differential interference contrast DIC microscopy[6,20], suggesting that ESVs were not formed at all in encysting Δcwp1 trophozoites. Correct condensation and partitioning of CWM in ESVs requires trafficking and cleavage of CWP2 by a cysteine proteinase[6,21,22]. To assess ESV formation in encysting Δcwp1 cells, we transfected line Δcwp1 with a previously tested expression vector (CWP2HA-BSR-pBS[6]) encoding a stage-regulated C'-terminally HA-tagged CWP2 reporter as a marker for CWM (Supplementary Fig. 3c). Subcellular distribution of CWP2HA as well as CWP1 was determined by IFA in encysting Δcwp1/pCWP1:CWP2HA cells and a transgenic control cell line WB/pCWP1:CWP2HA at 12–14 h p.I. CWP2HA distributed to distinctive post-ER ESV organelles of differentiating WB/pCWP1:CWP2HA trophozoites (Fig. 3a). However, the same reporter was not exported beyond the ER in Δcwp1/pCWP1:CWP2HA cells (Fig. 3a), as demonstrated by the signal overlap for deposition of CWP2HA and protein-disulfide isomerase (PDI) 2, a previously described ER marker detected by a specific antibody (Fig. 3b)[23]. Importantly, ESVs were not formed in encysting Δcwp1 cells and pseudocysts labelled with CWP2HA could not be identified. This was consistent with the absence of condensed ESVs in bright-field microscopy images of encysting Δcwp1/pCWP1:CWP2HA cells (Fig. 3a,b, DIC images). Taken together, the data demonstrate a fundamental role for CWP1 in export of CWP2 from the ER and formation of post-ER ESVs.

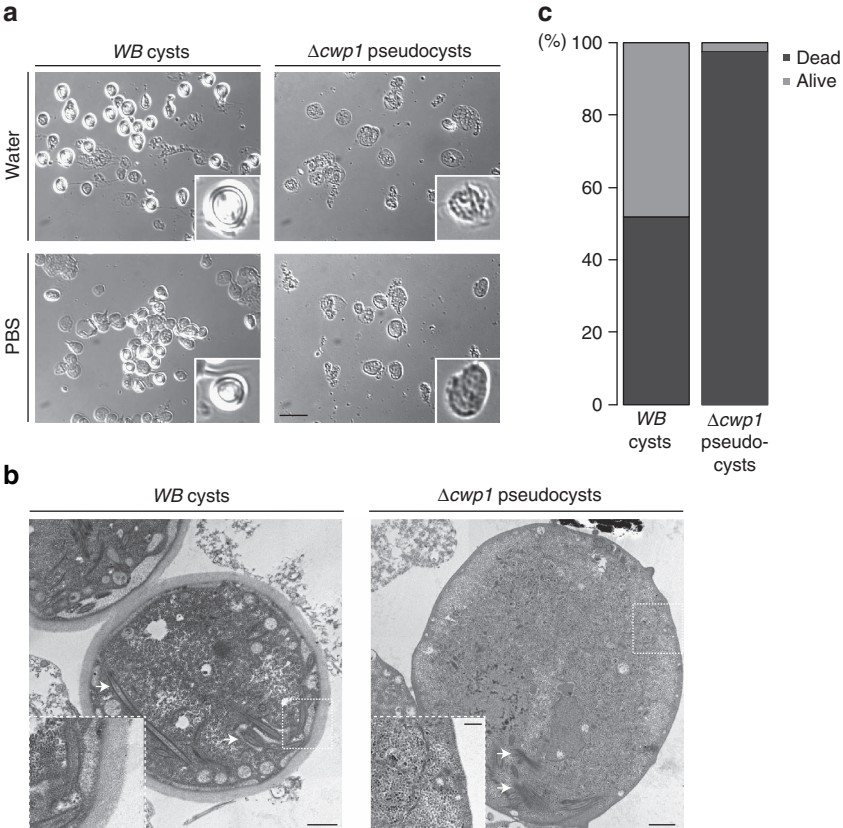

**Figure 2 | Δcwp1 pseudocysts lack a protective CW.** (a) Bright field microscopy: WB and Δcwp1-derived cysts/pseudocysts were tested for resistance to water (upper panels) and PBS (lower panels). WB cysts show refractile CWs; this optical effect is absent in Δcwp1 pseudocysts that appear damaged after exposure to water. Insets: individual cysts/pseudocysts. Scale bar: 25 μm. (b) tEM analysis: typical appearance of CWs (WB left panel); Δcwp1 pseudocysts lack a CW completely. Scale bar: 1 μm. Arrows indicate partially disassembled cytoskeletal elements. Insets scale bar: 0.2 μm. (c) AO/EB testing for cyst/pseudocyst viability highlights the fragility of cell culture-derived Δcwp1 pseudocysts compared with WB cysts. The y-axis indicates percentage.

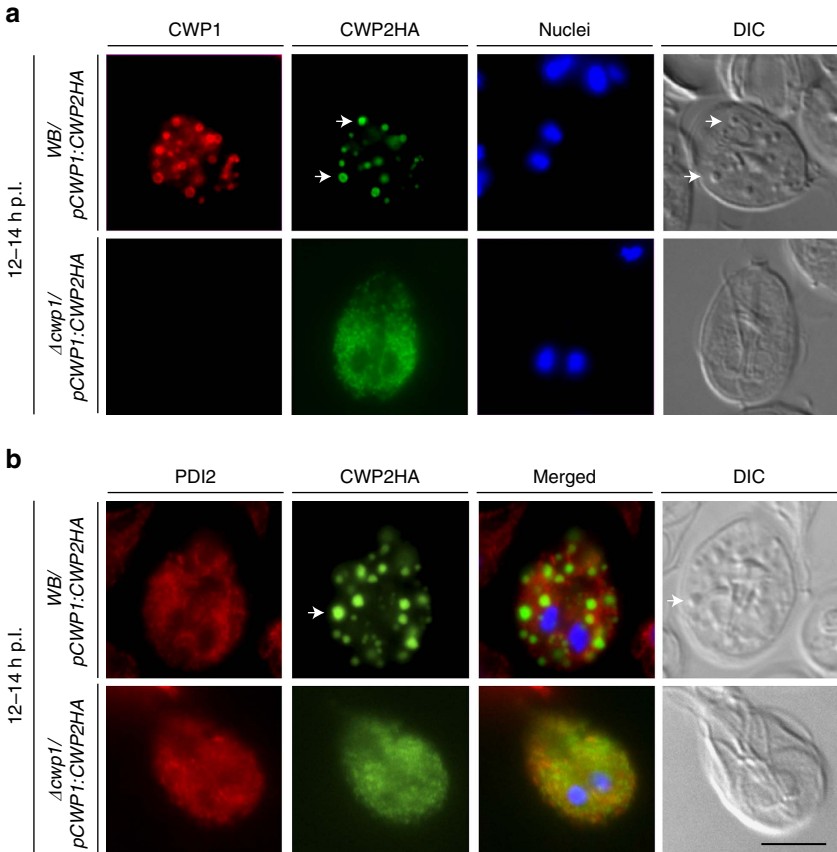

**Figure 3 | CWP1 ablation blocks ER export of a CWP2HA reporter in *Δcwp1/pCWP1:CWP2HA* cells.** IFA and bright field microscopy of encysting *WB/pCWP1:CWP2HA* and *Δcwp1/pCWP1:CWP2HA* cells at 12 h p.I. (**a**) ER export of a CWP2HA reporter in *WB/pCWP1:CWP2HA* cells and trafficking to nascent ESVs (arrows); CWP2HA remains blocked in the ER in *Δcwp1/pCWP1:CWP2HA* cells. (**b**) Export of the CWP2HA reporter to post-ER compartments (ESVs, green; arrow) in *WB/pCWP1:CWP2HA* cells (the ER marker PDI2 is labelled in red); PDI2 (red) and the CWP2HA reporter (green) co-distribute in the ER of *Δcwp1/pCWP1:CWP2HA* cells. Nuclear DNA (blue) stained with DAPI. DIC, differential interference contrast. Scale bar: 10 μm.

**Complementation of *Δcwp1* cells rescues ESV and CW formation.** To confirm that the trafficking phenotype in *Δcwp1* was exclusively due to the disruption of all *CWP1* alleles, we generated a genetically complemented cell line by transfecting *Δcwp1* trophozoites with an expression vector encoding a CWP1 expression cassette including a promoter as well as flanking 5′–3′ UTR sequences (construct CWP1-BSR-pBS; Supplementary Fig. 3c). Analysis of encysting *Δcwp1/pCWP1:CWP1* cells by IFA demonstrated expression and correct trafficking of CWP1 as in the *WB-C6* wild type control (Fig. 4). In line with previous reports[6,24], the CWP1 protein could be detected primarily in the ER and nascent ESVs of both *WB-C6* and *Δcwp1/pCWP1:CWP1* cells at 4 h p.I. (Fig. 4, 4 h p.I.). At 12 h p.I. correct formation of condensed ESVs was detected in the complemented line *Δcwp1/pCWP1:CWP1* (Fig. 4, 12 h p.I.)[6,24] and cysts derived from *Δcwp1/pCWP1:CWP1* cells were encased by a refractile CW that could be labelled with the anti-CWP1 mAb (Fig. 4, 24 h p.I.). Correct export of the CWP2HA reporter from the ER to ESVs (Fig. 4, 12 h p.I.) and the CW (Fig. 4, 24 h p.I.) was restored in a complemented background (line *Δcwp1/pCWP1:CWP1/pCWP1:CWP2HA*). Taken together, these data show that all defects associated with CWP1 ablation in line *Δcwp1* can be fully rescued by ectopic expression of a wild-type *CWP1* allele from an episomally maintained plasmid.

**_Δcwp1_ pseudocysts are able to capture free CWP1.** We showed that regulated ESV neogenesis and trafficking of the CW in encysting *Δcwp1* cells is abolished and pseudocysts are devoid of a protective extracellular matrix detectable by light and electron microscopy. However, because the unique β-1,3-linked GalNAc homopolymer of the CW is supposedly exported in non-ESV secretory organelles in *WB-C6* cells[25], the status of the glycan component in *Δcwp1* pseudocyst walls remained unknown. There is currently no specific agent for detection of the GalNAc homopolymer although the leucine-rich repeat domain of CWP1 was previously reported to be a lectin domain that binds specifically to the glycan in CWs[26]. Thus, native as well as recombinant soluble CWP1 can be used as a marker to detect the GalNAc homopolymer. Although tEM showed no evidence for glycan fibrils on pseudocysts, we used a CWP1 capture assay to address the question whether some of the CW glycan was exported to the surface of differentiated *Δcwp1* cells. Soluble CWP1 is released in significant amounts by encysting *WB-C6* cells, presumably due to incomplete cross-linking on the surface of newly formed cysts and can be harvested from cell culture supernatants[27]. Soluble CWP1 binds efficiently to deproteinated 'glycan cage' preparations of *WB-C6* cysts[26] but also to some extent to newly formed CWs[27]. Using the anti-CWP1 mAb we could detect captured CWP1 on unfixed *Δcwp1* pseudocysts incubated for 1 h in culture supernatant of encysting *WB-C6* cells (Fig. 5, upper left panel). No signal was obtained when *Δcwp1* pseudocysts were incubated with culture supernatant from non-induced *WB-C6* cells (Fig. 5, upper left panel). In addition, non-encysting *WB-C6* and *Δcwp1* trophozoites were unable to capture soluble CWP1 (Fig. 5, bottom panels), suggesting that

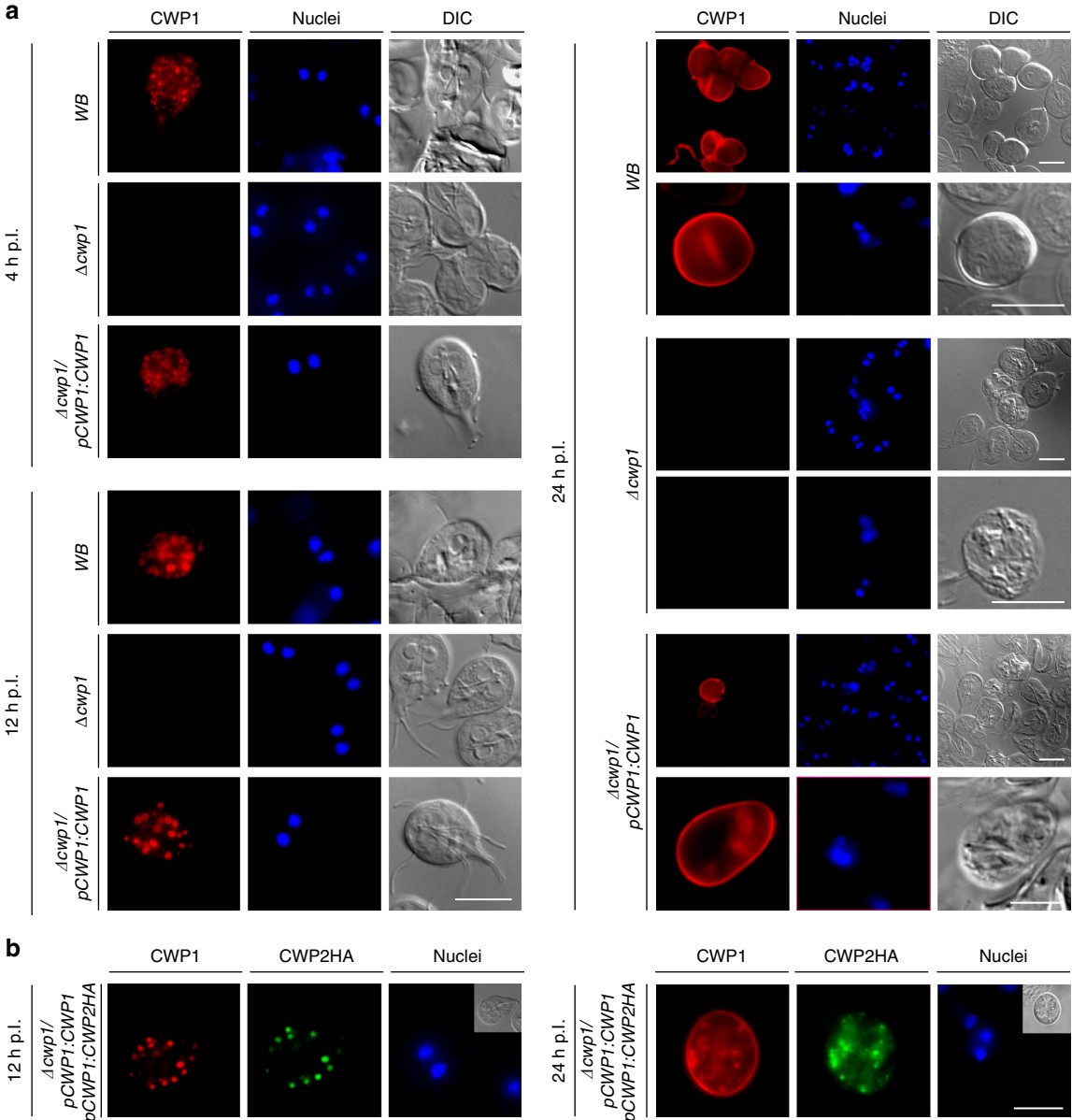

**Figure 4 | Genetic complementation restores ESV and cyst-wall formation in encysting *Δcwp1/pCWP1:CWP1* cells.** IFA and bright field microscopy of encysting WB (wild type control), *Δcwp1* (knock-out line) and *Δcwp1/pCWP1:CWP1* (complemented knock out) cells. (**a**) Expression and subcellular distribution of endogenous CWP1 at early (4 h p.l., top-left panel) and late (12 h p.l., bottom-left panel) stages of encystation and in cysts (24 h. p.l.). DIC: differential interference contrast. Scale bars: 10 μm. (**b**) Transfection of a wild-type *CWP1* expression vector (line *cwp1/pCWP1:CWP1/pCWP1:CWP2HA*) restores correct trafficking of a CWP2HA reporter via ESVs (left panel) to the nascent CW (right panel). Nuclei (blue) were stained with DAPI. Insets: DIC images. Hours (h) p.l.: hours post induction of encystation. Scale bars: 10 μm.

CWP1 binds specifically to the surface of *WB-C6* cysts or *Δcwp1* pseudocysts. This is indirect evidence that *Δcwp1* pseudocysts might still secrete CW glycan to the surface despite the complete abolishment of CWP trafficking and absence of ESVs.

## Discussion

*G. lamblia* is an intestinal protozoan parasite with a strict requirement for differentiation to a cyst stage and back to a trophozoite for transmission to a new host. Genetic manipulation of *Giardia* has been established for over two decades with first attempts by electroporation and selection of circular plasmid episomes[28], followed by strategies using *Giardia* virus RNA as a vector[29,30], and finally integration of linear vector DNA for modification of chromosomal DNA[14,31,32]. Recently, genome

engineering using the Cre/*loxP* system has allowed elimination of a single allele followed by the removal of the selection marker from the chromosomal DNA[11]. In the attempt to perform functional characterizations of target genes, tools for depletion of target mRNA stability and/or inhibition of translation were developed and adapted to *Giardia*[33,34]. It is important to note that these approaches cannot ensure absence of any given protein since mRNA depletion was reported at anywhere between 22 and 70% (refs 35–38), with the strong likelihood of ambiguous results in phenotypic analyses. In addition to expression control, ectopic expression of dominant-negative variants of GTPases was used in some instances to disrupt protein function and elicit quantifiable phenotypes[13,39,40]. The availability of only two highly effective antibiotic resistance markers for selection of transgenes[31] and the organization of the genomic DNA of trophozoites in two diploid

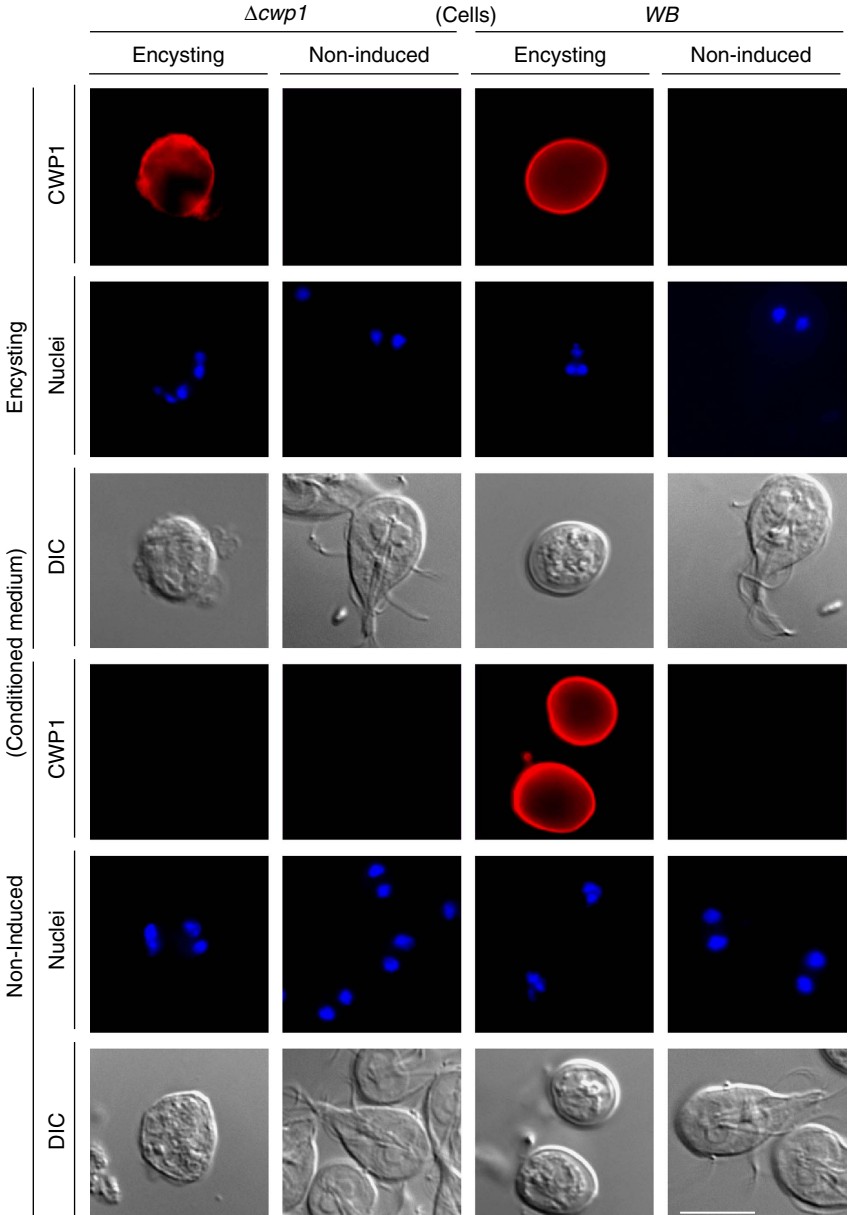

**Figure 5 | The surface of _Δcwp1_ pseudocysts can capture soluble CWP1.** IFA analysis of _Δcwp1_ pseudocysts incubated with conditioned medium either from encysting (top, left panel) or non-encysting (top, right panel) _WB_ cultures. Soluble CWP1 was captured on the surface of _Δcwp1_ pseudocysts only after incubation with conditioned medium from encysting WB cells but not trophozoites (left panels). WB cysts (control) react with the anti-CWP1 antibody (right panel). _Δcwp1_ or WB trophozoites are unable to capture soluble CWP1 from conditioned medium. Nuclei (blue) were stained with DAPI. DIC, differential interference contrast. Scale bar: 10 μm.

nuclei has made engineering of _G. lamblia_ knock-out lines by ablation of all four target gene alleles very challenging. Here, we report on how we addressed this limitation using a method for serial locus disruption based on the Cre/loxP system. By targeting CWP1, we took advantage of the strict stage-specific regulation of CWP1 expression, in addition to established markers and morphological landmarks associated with the entire process of cyst formation[6,12,41] for the analysis of a _Δcwp1_ phenotype. Together with _Trypanosoma brucei_[42], _G. lamblia_ is the only other species within the Excavata supergroup in which genome editing was performed to address questions on pathogen transmission.

We addressed the paucity of reliable selection markers in _Giardia_ by using sequential allelic disruption combined with subsequent excision of the antibiotic-resistance expression cassette. Iteration of the gene disruption events was necessary

because, as previously indicated[14], only one allele is targeted for replacement _per_ transfection event with linearized DNA, although multiple circular plasmids expressing different genes of interest can be maintained in a single transgenic cell[43]. In addition, transfecting trophozoites with either a circular episomally maintained or linear integrated vector targets only one of the two nuclei[44]. This could mean that one nucleus is _a priori_ refractive to integrating foreign DNA, which would add another level of difficulty for obtaining a full knockout. However, we have not seen any evidence for this, that is, knock-out of the third CWP1 allele was possible without any changes in the protocol. Although our data point to its effectiveness, the sequential knock-out strategy could be further improved by addressing two main bottlenecks, concerning (a) loss of Cre-encoding plasmid and (b) efficiency of Cre/loxP mediated excision. Unexpectedly,

the complete loss of transfected plasmids such as the Cre-encoding episome required propagation in culture without selective pressure of up to six weeks. This is all the more surprising given that none of the constructs used in this study carry specific genetic information for replication and maintenance in *Giardia*, thereby raising the question as to how these genetic elements are maintained *in vitro* within a population, apparently without any influence on fitness. This aspect of exogenous DNA maintenance in *G. lamblia* is relevant to our methodology in that selection of a knock-out event by ensuring genomic insertion of a functional puromycin-resistance expression cassette can only be iterated in the absence of Cre. Although no negative selection systems are yet established in *Giardia*, it may be possible to adapt methods such as the thymidine kinase-based strategy developed in *T. brucei*[45], thus accelerating loss of redundant episomal elements. In addition, the use of chimeric *loxP* sites generating incompatible derivatives[46] would drive the Cre-maintained equilibrium towards excision of the 'floxed' puromycin-resistance cassette. As an alternative to sequential targeted gene disruption we also tested the feasibility of implementing CRISPR-Cas9 genome editing systems in *G. lamblia*. However, despite the engineering of functional nuclear localization signals, the full-length *Streptococcus pyogenes* Cas9 enzyme expressed in trophozoites was excluded from the nucleus, while a truncated albeit inactive variant was efficiently targeted to both nuclei (Supplementary Fig. 4). This suggests that primary and/or secondary structure features of the Cas9 protein, which is fully functional in other protozoan parasites, for example, *Toxoplasma gondii*, are incompatible with nuclear import in *Giardia*. Further research is necessary to test whether this genome editing approach can be implemented also in diplomonads.

Differentiating *Δcwp1* cells fail to develop ESVs and are unable to traffic CWP2 further than the ER. Because the CWP2 marker is abundant in the ER in early encysting cells and cannot exit the ER, the most likely explanation is that the blocked protein is degraded in the 14–18 h between peak production[4,47] and until *Δcwp1* pseudocysts were harvested and labelled for immunofluorescence. We have established previously that CWPs are exported from ER exit sites via a Sar1-dependent trafficking pathway into growing ESVs before post translational processing, partitioning, and sorting of the CWM occurs[6,12,13]. How this cargo is selected for export from the ER remains unknown. The data provide the first indication that CWP export from the ER is a cooperative process requiring the presence of all components. In line with this, overexpression of CWP2HA was not sufficient to rescue the pseudocyst phenotype, thereby showing that CWPs are not functionally redundant. Thus, the new genome editing tools developed herein will allow addressing the question whether encysting cells use cooperative ER export combined with degradation of CWPs that are unable to exit the ER to ensure trafficking of the required amount and ratios of CWPs for building the CW.

Although *Δcwp1* cells are unable to establish the regulated trafficking pathway for CWM export, these cells nevertheless respond to the encystation stimulus by undergoing morphological differentiation to oval-shaped pseudocysts including cytoskeleton remodeling, that is, resorption of flagella and disassembly of the ventral disk. Of note, pseudocysts also have two nuclear pairs, a hallmark of cysts. In WB cells nuclear division occurs simultaneously with secretion and polymerization of the outer CW. This initially suggested coordination of secretion and cell cycle progression. However, we later showed that blocking secretion of the CWM by conditional expression of a dominant-negative Arf1 GTPase[13] can decouple these two processes. Induced cells expressing the mutant Arf1 are able to accumulate ESVs but fail to secrete CWPs and produce wall-less pseudocysts, whereas ESV biogenesis in *Δcwp1* is abolished entirely. This is also the case in induced trophozoites expressing a mutated Sar1-GTPase, which blocks protein export from the ER completely[12]. However, in this cell line, pseudocysts never develop, suggesting that morphological differentiation and cytoskeleton remodelling are independent of CWP trafficking but still require functional ER export machinery.

The β-1,3-linked glycan homopolymer is synthesized by a set of upregulated enzymes in encysting trophozoites and comprises >60% of the CW matrix[48]. Although postulated to be trafficked in dedicated transport vesicles[25,26], the details of CW glycan trafficking and integration with CWPs to form a biopolymer remain unclear. Complete deproteination of matured WB CWs exposes a fibrillary mesh of long unbranched glycan chains in tEM micrographs. These glycan 'cages' efficiently capture soluble CWP1, which led to the conclusion that these proteins are the only known lectins that are able to recognize this β-1,3 linkage[26]. Although tEM provided no evidence that *Δcwp1* pseudocysts exposed glycan on the PM, we tested this by using its ability to capture soluble CWP1. Indeed, similar to walls of mature WB cysts, *Δcwp1* pseudocysts are distinctly labelled with soluble CWP1 detected with the anti-CWP1 mAb. This is not conclusive evidence for the presence of the CW glycan on the surface of pseudocysts, but the finding that soluble CWP1 does not bind to trophozoites suggests that surface remodelling is an additional facet of pseudocyst formation. This is consistent with the idea that the CW glycan is exported independently of ESVs. Nevertheless, complete unravelling of glycan trafficking and integration with the CWPs as a complex, mechanically and chemically resistant polymer will require development of a heterologous-specific label for this GalNAc polymer, which is unique to *G. lamblia*.

## Methods
All sequences for oligonucleotide primers used in this work are listed in Supplementary Table 1.

**Giardia cell culture for encystation and transfection.** *G. lamblia* WB-C6 (ATCC catalog number 50803) trophozoites were grown and harvested using standard protocols[49]. Transgenic parasites were generated according to established protocols by electroporation of linearized or circular plasmid vectors prepared from *E. coli* as described in ref. 4. Encystation was induced using either the two-step method involving incubation in bile-free pre-encystation medium, or medium prepared using lipid-depleted serum[4,13]. The 'high bile' method for encystation was used to maximize cyst production[18]. After selection for puromycin (final conc. 50 µg ml⁻¹), neomycin (G418; final conc. 150 µg ml⁻¹) or blasticidin (final conc. 75 µg ml⁻¹; ref. 31) resistance, transgenic *G. lamblia* cell lines were cultured and analysed without antibiotic. Lines used in this work: WB-C6 (WB), *Δcwp1* (*Δcwp1-1::loxP*, *Δcwp1-2::lox5171*, *cwp1-3Δ::PAC*, *cwp1-4::NEO*), *Δcwp1/pCWP1:CWP1*, *Δcwp1/pCWP1:CWP2HA*, WB /pCWP1:CWP2HA and *Δcwp1/pCWP1:CWP1/pCWP1:CWP2HA*.

**CWP1KO1-4 plasmid synthesis and engineering of line Δcwp1.** Line *Δcwp1* (*Δcwp1-1::loxP*, *Δcwp1-2::lox5171*, *Δcwp1-3::PAC*, *cwp1-4::NEO*) was generated in a WB-C6 genetic background based on the sequential deletion or interruption of all four alleles coding for CWP1. To do this, each locus was targeted for homologous recombination with linearized constructs CWP1KO1 to CWP1KO4 (Supplementary Fig. 3a). These feature a puromycin-resistance expression cassette (*PAC*) driven by a constitutively-active promoter flanked by two *loxP* sites in the same orientation[11]. Upstream and downstream of this 'floxed' *PAC* cassette we cloned ca. 1,000 bp-long genomic DNA stretches retrieved from the Giardia Genome database (www.giardiaDB.org), upstream and downstream of the *cwp1* locus, respectively. To obtain sequential allelic knock-out without interfering with preceding deletion events, constructs CWP1KO1 to CWP1KO4 were engineered in a 'nested' configuration (Supplementary Fig. 3a). Furthermore, constructs CWP1KO2 and CWP1KO3 carry *lox* pairs presenting slight sequence modifications compared to wild-type *loxP* sites that drive recombination within the pair, without engaging with residual *lox* sites[50]. Following selection of cells resistant to puromycin, these were then transfected with a constitutively Cre-encoding plasmid to mediate excision of the floxed *PAC* cassette (Supplementary Fig. 3b). Selection for this construct with neomycin was stopped after the first passage to

promote loss of the plasmid from the transfected population (on average after 5 weeks). Given that Cre recombinase can both excise and re-integrate loxP-carrying DNA fragments[51], single-cell plating in 96-well plates and testing for puromycin resistance were then performed to raise clonal lines where the floxed cassette had been entirely excised from the genome. Following confirmation of their clonal status by PCR analysis of genomic (g) DNA, selected lines were expanded in culture. This procedure was repeated twice to integrate constructs CWP1KO2 and CWP1KO3, thereby achieving knock-out of 3 out of 4 CWP1 alleles. We disrupted the remaining allele with construct CWP1KO4 by inserting an inverted neomycin-resistance expression cassette (NEO) 227 bp downstream of the start codon for the CWP1 locus. This resulted in the first engineered knock-out cell line in G. lamblia with three ablated and one interrupted CWP1 alleles (Fig. 1a). This mutant line appears to grow normally, with doubling times similar to wild-type non-transgenic WB-C6 cells.

**Plasmids for complementation tests and Cas9 expression.** Constructs CWP1-BSR-pBS and CWP2HA-BSR-pBS were cloned in a pBluescript SK+ background (Stratagene) containing a blasticidin-S-deaminase expression cassette (BSR) for blasticidin-based selection (Supplementary Fig. 3c)[52]. Both constructs carry CWP1 (CWP1-BSR-pBS) or C-terminally HA tagged CWP2 (CWP2HA-BSR-pBS) open reading frames driven by a minimal CWP1 promoter. Recombinant full length (4,254 bp) and shortened (1,044 bp) variants of the S. pyogenes Cas9 open reading frame were amplified from the pSAG1_CAS9-U6_sgUPRT construct[53] to include a 5′ tandem nuclear localization signal derived from the SV-40 Large T antigen[54] to yield constructs pSec-tNLS-Cas9fl-HA and pSec-tNLS-Cas9s-HA, respectively. Expression from both constructs is regulated by the predicted minimal G. lamblia Sec23 promoter (open reading frame GL50803_9376)[12]. Cloning and construct propagation were performed according to standard protocols.

**Genomic DNA and 3′RACE analysis.** PCR analysis of gDNA was performed as previously reported[11]. 3′RACE (rapid amplification of cDNA 3′ ends) was done as reported previously[4]. In brief, total RNA was extracted from ca. 20 million trophozoites using TriZol (InVitrogen) and mRNA was specifically reverse-transcribed using the Superscript III First-strand synthesis kit (ThermoFisher) according to the manufacturer's indications. The resulting cDNA was amplified in standard conditions. All sequences of oligonucleotide primers used in this section are listed in Supplementary Table 1.

**Total protein extraction and immunoblot analysis.** The equivalent of $8 \times 10^7$ cells was resuspended in 0.5 ml of 1X Laemmli buffer, incubated at room temperature for 30 min and then cold-centrifuged for 15 min at $16,000 \times g$. The supernatant was separated, supplemented with 10 mM DTT and boiled for 5 min. After a second centrifugation, the supernatant was loaded and run on denaturing SDS-PAGE gels (4%/10%), followed by immunoblotting in standard conditions (Supplementary Fig. 5). Blots were probed with anti-CWP1 antibody (dilution 1:1,000; product no. A300 TR-R-20x, Waterborne Inc.) followed by anti-mouse horseradish peroxidase-coupled antibody (dilution 1:5000; product no. NB7539, Novus Biologicals) and developed using a chemiluminescent substrate (Westernbright; Thermofisher Scientific). Gels for loading controls were stained with Instant Blue Coomassie stain (Expedeon).

**Immunofluorescence analysis (IFA) and microscopy.** Preparation of chemically fixed cells for immunofluorescence and analysis of subcellular distribution of reporter proteins by wide-field microscopy were done as described previously[6,13]. CWP1 labelling was performed using a Texas-Red conjugated monoclonal antibody (dilution 1:80; product no. A300 TR-R-20x,Waterborne Inc.). The HA epitope tag was detected using a fluorescein isothiocyanate (FITC)-conjugated monoclonal antibody (dilution 1:50; clone 3F10, Roche). Giardia PDI2 was detected using a self-made anti-PDI2 antibody (dilution 1:1,000 (ref. 23)) followed by an anti-mouse-Alexa594 secondary antibody (dilution 1:300; product no. A-11032, Molecular Probes). Nuclei were labelled with 4′,6-diamidino-2-phenylindole (DAPI).

**Viability test with acridine orange and ethidium bromide.** Freshly harvested cysts were stained with acridine orange and ethidium bromide both at a final concentration of 100 µg ml⁻¹ in PBS[19], observed under a wide-field fluorescence microscope and scored for viability (>150 cysts per sample). Cysts were not exposed to water prior to staining.

**Transmission electron microscopy of ultrathin sections.** Cell pellets were re-suspended with 2.5% glutaraldehyde in 0.1 M Na/K-phosphate, transferred to a microtube and centrifuged for 20 min at 3400 Xg. The pellets were rinsed with 0.1 M Na/K-phosphate and postfixed with 1% osmium tetroxide in 0.1 M Na/K-phosphate for 1 h, dehydrated in a series of ethanol starting at 70%, and after transferring into acetone embedded in epon resin followed by polymerization at 60 °C for 2.5 days. Sections of 60–80 nm thickness were stained with uranyl acetate

and lead citrate, and analysed in a transmission electron microscope (CM12, FEI, Eindhoven, The Netherlands) equipped with a CCD camera (Ultrascan 1,000, Gatan, Pleasanton, CA, USA) at an acceleration voltage of 100 kV.

**Data availability.** The authors declare that all data supporting the findings of this study are available within the paper and its Supplementary Information files, or are available from the authors upon request.

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

## Acknowledgements

We thank Ms Therese Michel for technical assistance and Dr Petra Wampfler for helpful discussions. Mr Asgeir Astvaldsson and Prof Staffan Svärd are gratefully acknowledged for sharing the construct carrying a blasticidin-resistance expression cassette. We are grateful for advice on the anti-CWP1 mAb from Drs Hugo Lujan, Theodore Nash and Henry Stibbs. This study was funded by grants from the Swiss National Science Foundation #31003A_140803 and #31003A_166437.

## Author contributions

J.A.E., S.D.H., A.B.H. and C.F. designed and executed experiments. E.M.S. performed the electron microscopy analysis. J.A.E. made all figures. A.B.H. and C.F. wrote the manuscript.

## Additional information

**Competing financial interests:** The authors declare no competing financial interests.

**Publisher's note**: 

