## [Peer Review File · Nature Communications]

Reviewers' comments:

Reviewer #1 (Remarks to the Author):

A major difficulty in investigations of *Giardia lamblia*, the binucleate parasite that causes diarrhea, is knocking out genes, which are each present in four copies. In this well-written and exciting paper, the authors, who are experts in the cell biology of *Giardia* and its genetic manipulation, perform a "tour de knock-out," using Cre/loxP system to eliminate all four copies of the *cwp1* gene, which encodes the most abundant cyst wall protein. They show that Δ CWP1 parasites are unable to form a cyst wall, when parasites are induced to encyst in culture. Further they show that encystation-specific vesicles containing another abundant cyst wall protein (CWP2) do not form properly. They use an indirect assay using medium from encysting parasites as a source of native CWP1 to suggest that the abundant sugar polymer in the cyst wall, which is composed of β -1,3-linked GalNAc, is present on the surface of encysting Δ CWP1 parasites. Finally, they show that the knockout is fully complemented by an episomally maintained CWP1 expression vector.

This is a high impact genetic experiment, particularly because CRISPR/Cas9 did not work in their hands and has not yet been reported by others. Hearty congratulations to the authors! I have three suggestions that might improve an excellent paper.

1. While most of the figures are clear and conservatively interpreted, the authors might be cautioned to say too much about the "pseudocysts" formed by Δ CWP1 parasites, because they are 98% dead. Indeed one needs to be cautious interpreting wild-type organisms encysting in vitro, half of which are dead!
2. They should use the MBP-CWP1 fusion protein, which binds to the β -1,3-linked GalNAc polymer (ref. 22), to determine what happens to vesicles containing the β -1,3-linked GalNAc polymer in the Δ CWP1 parasites. Their indirect assay using medium from encysting parasites as a source of native CWP1 did not answer this question.
3. The complementation experiment provides a great opportunity to test in encysting parasites the two components of CWP1, which are a leucine-rich repeat domain and a Cys-rich domain (again ref. 22). While the LRR domain of CWP1 appears to be a lectin that binds the β -1,3-linked GalNAc polymer, the Cys-rich domain remains uncharacterized. It would be of great interest then to determine the phenotype of the complementation with the CWP1 LRR domain without the Cys-rich domain.

Reviewer #2 (Remarks to the Author):

The manuscript entitled "A fundamental role for Cyst Wall Protein 1 in neogenesis of Golgi-like organelles and cyst wall biosynthesis in *Giardia lamblia*" attempts to show that total abrogation of the *cwp1* gene is essential for the development of Encystation-specific Secretory Vesicles (ESVs) and assembly of the cyst wall in this important parasite. Just the

title is ambiguous since they only focused on CWP1 and abolishing expression of other CWPs may also be "fundamental".

The authors report, for the first time in *Giardia*, the complete knock out of the *cwp1* gene in the two diploid nuclei of the parasite by using a sequential Cre/loxP-based approach and selection marker rescue strategies. The results of this experimental approach appear effective in allowing genomic editing. However, this sequential approach has not been analyzed in detail after each step, which makes the procedure difficult to follow. The authors should include an analysis of the expression of target gene after each round of the process to verify that the system works and, at the same time, to observe the effects of single allele disruption on the physiology of the parasite during cyst formation.

The authors performed 3'-RACE and RT-PCR to probe the disappearance of the *cwp1* transcript (It is confusing. See below.). However, the methods designed to confirm this experimental evidence look insufficient. Why the authors do not use Western blotting to test for the total disappearance of CWP1 is unclear.

The use in immunofluorescence assays of a commercial antibody against CWP1 for Waterborn Inc., which has not been published by the company, is an important weakness of this work. The original antibody developed against a 65 kDa cyst wall product by Stibbs et al. in the early 1990's was used to clone the *cwp1* gene encoding a 26 kDa protein by Mowatt et al. in 1995, but the current antibody that the company sells is not longer the original 5C1 monoclonal antibody. For that reason, a full characterization of this reagent is needed before use in Western blotting and IF. On the other hand, fully characterized monoclonal antibodies against CWP1 and CWP2 have been generated and reported by several groups (Lujan, Faubert, Nash, etc.). Why the authors have not used those antibodies is unclear. TEM studies in which immuno EM with anti-CWPs antibodies look very feasible and necessary.

In this scenario, knocking out CWP2, alone or jointly with CWP1, would provide better descriptions of the molecular basis of cyst wall formation in *Giardia*. So far, no antibody to CWP3 has been reported. Again, although the strategy developed by the authors to knock out genes for the first time in this polyploid, binucleate parasite is of outstanding relevance in the field, the lack of appropriate controls require taking this technique with caution, at least for the moment.

Another important issue regarding this work is the fact that the authors claim that secretory granules that form de novo from the ER during trophozoite encystation are Golgi-like structures. Reports from several groups showed that the molecular chaperone BiP (Gottig et al. 2006; Touz et al. 2002) and the ER marker Yip1 (Stefanic et al., 2006; Wampfer et al., 2014) are present in these granules, that these granules do not contain the KDEL-receptor (Gottig et al. 2006; Elias et al. 2008), among other Golgi markers (Stefanic et al., 2006; Wampfer et al., 2014), indicating that the ESVs are generated from the ER by condensation of CWPs and specific sorting events (Gottig et al., 2006). This work is not the first evidence that ESV formation from the ER requires the presence of several molecules working in conjunction (Gottig et al., 2006; Touz et al. 2002, 2003), as stated in the discussion

section. How can ER resident proteins be in the ESVs if they are Golgi-like organelles? How do the Golgi-like organelles lack more specific Golgi markers? Gottig et al. (2006) have shown that a basic c-terminal extension only present in CWP2 is necessary for ESV biogenesis from the ER. Thus, knocking out CWP2 appears as the most relevant experiment to study ESV formation in Giardia and all other comments are speculative in essence.

Additionally, the detection of other cyst wall components in the so called "pseudocysts" devoid of CWP1 should be tested. For that, the available antibodies to untagged CWPs may provide unequivocal characterization of these structures and avoid speculations regarding Golgi characteristics. Previous reports from Hehl's group have shown that different tagged versions of CWPs have distinct roles in cyst wall formation (Faso & Hehl, 2011). However, the use of untagged but detectable constructs seems essential since the tags can modify protein behavior, in particular during the dynamic process of Giardia encystation. Moreover, the authors claim that the "pseudocysts structures" are similar to that previously shown by Hehl's group when disturbing ARF1 function. However, ARF1 is also present in the ER and in the lysosomal/endosomal compartment of Giardia (Lujan et al., 1995). Therefore, what is the link between ARF1, ESVs and CWPs? A clear discussion about the situation is needed.

The authors also stated that, citing recent studies of Hehl's group (Konrad et al., 2010), correct condensation of CWPs is necessary for segregation into the ESVs. But that was proposed earlier by Gottig et al. in 2006. Moreover, given that previous reports showed important difference in the interpretation of the results between these two groups, the authors of this manuscript should comment on the discrepancies and perform additional experiments to provide new insight into this scientific disagreement.

The authors also claim that CWP1 is released to the culture medium and that binds the carbohydrate polymer of the cyst wall, but what about CWP2 (or even CWP3). The minor differences between the LLR domains of these proteins do not guarantee that the only one tested is the only one working (Chatterjee et al.; 2010). No comparative study between the LLR of CWPs has been reported.

Another important consideration regarding this work is the use of episomal vectors to rescue the knock out phenotype. In most protozoa, expression of a given protein is not universal with episomal vectors. Thus, how do the authors observe a complete rescue of the pseudocyst phenotype?

The authors also refer to a work of Poxleitner et al.; 2008, in which by using either linearized DNA or episomal vectors only one nucleus of the parasite is targeted. However, they do not need to modify their protocol to have successful results. On one hand, they stated that the efficiency of their strategy needs further improvements and, on the other hand, they claim complete success. It needs to be clarified.

From the 47 references of the manuscript, almost a half of them are from the Hehl's group while results and hypothesis reported by other groups are completely neglected. The authors should acknowledge the work of others and seriously discuss their scientific differences.

Figure 1C: It is confusing since in the text the authors indicated that they perform 3'-RACE but in the figure legend they show an RT-PCR analysis. The primer 1040 is for RACE, as indicated in Table 1 as k-adaptor. This can be easily clarified but the problem is that the CWP1 transcript is present in NON-ENCYSTING and PRE-ENCYSTING trophozoites, when they stated in the Abstract "CWP1 is expressed only in encysting cells"; in line 40 "the genes coding for CWPs are completely silenced in trophozoite"; and in line 277 "By targeting CWP1, we took advantage of the STRICT stage-specific regulation of CWP1 expression". These contradictions are objectionable.

In Fig. S2, the results of amplification of primers pairs 2414/2415 and 2416/2417 are missing. Please clarify why.

Finally, the authors stated that the CRISPR/Cas9 system does not work in Giardia based on their failure to edit the parasite genome using that valuable tool. Many sentences in the abstract, results and discussion sections (in addition to several references) are based on their negative results (not shown). It is unclear why the authors reached their conclusion based on a technique that is not described at all in the manuscript. It is not necessary to elaborate on a technique that did not work in the authors' hands to make their editing approach more important. If their system works as they claim, why discourage other groups to attempt to make CRISPR/Cas9 to work in Giardia.

Reviewer #3 (Remarks to the Author):

The authors employed the Cre/loxP methodology, which they previously developed for use in Giardia, to knockout all 4 genes of CWP1 to explore the loss of this gene on the encystation, ESV formation and cyst wall development. Although the authors did not develop this methodology, they cleverly applied it for use in Giardia. This work took an enormous amount of work and diligence, and it is novel and of interest to cellular biologists working in secretion, Golgi genesis and function, encystation and persons working in Giardia and other protozoa that encyst. Because Giardia is so unique in some aspects of its biology noted above, it is unclear how general the findings will apply to other organisms. One of the most interesting aspects of Giardia biology is the need for and functioning of two supposedly equally functioning nuclei. This system has the capability to knockdown both genes in only one nucleus so that the need and function of each nucleus, how they coordinate with each other and other interesting aspects of this bi-nucleated protozoa can be studied. One only needs to add epitope markers to tag each allele. In Giardia an easy and sometimes very effective method to decrease and even eliminate transcription is expression of antisense transcripts to the gene of interest. Levels can be undetectable. There was some discussion in an earlier paper from the same lab originally describing the Cre/LoxP system in Giardia concerning the limitations of antisense methodology, but it may be very effective and save a lot of work and give the same answer. Perhaps the prior discussion can be repeated. Obviously Cre/loxP has a number of advantages, but as seen in

this work, it demanding and labor intensive. The biological effects of CWP1 knockout are interesting and well proven. However, the ins and outs of Golgi formation, cyst wall protein interactions, ESV formation and cyst wall protein formation will be hard sledding for the most readers.

The paper is well presented and written. It appears to be technically correct and the Giardia methods clear and correct. In the testing of Δ cwp1 cysts for viability using impermeable dyes, clarify if the cysts tested with were exposed to water first or not. In the discussion of antibiotic resistance genes for Giardia (line274), there is mention of only two selectable antibiotic systems for Giardia but are there not three, blasticidin, puromycin and G418- all actually used in this work? The history of gene introduction and manipulation in Giardia starting on line 264 is not correct. The first published work was by Yee et al in June 1995. They used a plasmid containing luciferase with 5' and 3' Giardia GDH. Shortly thereafter in September of the same year Yu et al published a Giardia RNA virus construct; they added a neomycin selectable marker in 1996. There is no mention of antisense methodology by Touz et al first used in 2002. Singer et al and Sun simultaneously published on a stable expression vectors etc. but only Sun is referenced.

Response to reviewers for manuscript NCOMMS-16-16996

Reviewers' comments:

Reviewer #1 (Remarks to the Author):

A major difficulty in investigations of *Giardia lamblia*, the binucleate parasite that causes diarrhea, is knocking out genes, which are each present in four copies. In this well-written and exciting paper, the authors, who are experts in the cell biology of *Giardia* and its genetic manipulation, perform a “tour de knock-out,” using Cre/loxP system to eliminate all four copies of the *cwp1* gene, which encodes the most abundant cyst wall protein. They show that Δ CWP1 parasites are unable to form a cyst wall, when parasites are induced to encyst in culture. Further they show that encystation-specific vesicles containing another abundant cyst wall protein (CWP2) do not form properly. They use an indirect assay using medium from encysting parasites as a source of native CWP1 to suggest that the abundant sugar polymer in the cyst wall, which is composed of β -1,3-linked GalNAc, is present on the surface of encysting Δ CWP1 parasites. Finally, they show that the knockout is fully complemented by an episomally maintained CWP1 expression vector.

This is a high impact genetic experiment, particularly because CRISPR/Cas9 did not work in their hands and has not yet been reported by others. Hearty congratulations to the authors! I have three suggestions that might improve an excellent paper.

> *We thank the reviewer for their appreciation of our work.*

1. While most of the figures are clear and conservatively interpreted, the authors might be cautioned to say too much about the “pseudocysts” formed by Δ CWP1 parasites, because they are 98% dead. Indeed one needs to be cautious interpreting wild-type organisms encysting in vitro, half of which are dead!

> *We thank the Reviewer for this comment – indeed, the appropriate designation to use for the wall-less Δ CWP1 cells which represent a developmental endpoint after induction to differentiate was the subject of much discussion in our group. We finally decided on “pseudocyst” as the compromise which reflects the phenotype most accurately. We reasoned that although pseudocysts present gross morphological aberrations and do not have a proper cyst wall with respect to their wildtype counterparts, these structures are still quadrinucleated and have disassembled flagella and the ventral disk attachment organelle which are both hallmarks of the completed stage-differentiation process giving rise to cysts. For these reasons and also for want of a more accurate term we decided to use “pseudocyst”.*

2. They should use the MBP-CWP1 fusion protein, which binds to the β -1,3-linked GalNAc polymer (ref. 22), to determine what happens to vesicles containing the β -1,3-linked GalNAc polymer in the Δ CWP1 parasites. Their indirect assay using medium from encysting parasites as a source of native CWP1 did not answer this question.

> *We agree with the Reviewer that investigation of GalNAc polymer trafficking is an interesting question which can be addressed in a Δ CWP1 context. However, it's important to note that the work presented here is focused on the extent of surface remodelling, specifically*

whether glycan is present on the surface of pseudocysts. To address this question, we deliberately chose to use the native, correctly folded Giardia protein as a reporter to avoid the possibility of artefacts produced by the use of unfolded recombinant protein isolated from bacteria. However, since the presence of carbohydrate-positive vesicles remains undetermined in Δ CWP1 encysting cells, we have amended line 233 to “the status of the glycan component in Δ cwp1 pseudocyst walls remained unknown” to focus exclusively on pseudocysts.

3. The complementation experiment provides a great opportunity to test in encysting parasites the two components of CWP1, which are a leucine-rich repeat domain and a Cys-rich domain (again ref. 22). While the LRR domain of CWP1 appears to be a lectin that binds the β -1,3-linked GalNAc polymer, the Cys-rich domain remains uncharacterized. It would be of great interest then to determine the phenotype of the complementation with the CWP1 LRR domain without the Cys-rich domain.

> We fully agree with the reviewer that a detailed structure-function analysis of CWPs, i.e. complementation tests using individual CWP1 domains will allow for a thorough dissection of the contribution of each domain to overall CWP1 function. These extensive sets of experiments will be underway as soon as we have overcome a significant technical hurdle, namely the development of a specific exogenous marker for the unique β -1,3-linked GalNAc polymer. We plan to publish the results of this large separate study in combination with the characterization of Δ CWP2 and Δ CWP3 phenotypes.

Reviewer #2 (Remarks to the Author):

The manuscript entitled “A fundamental role for Cyst Wall Protein 1 in neogenesis of Golgi-like organelles and cyst wall biosynthesis in Giardia lamblia” attempts to show that total abrogation of the cwp1 gene is essential for the development of Encystation-specific Secretory Vesicles (ESVs) and assembly of the cyst wall in this important parasite. Just the title is ambiguous since they only focused on CWP1 and abolishing expression of other CWPs may also be “fundamental”.

> Based on the severity of the Δ CWP1 phenotype, we can confidently assign a fundamental role to CWP1 as a component of central importance in ESV biogenesis and hence cyst wall deposition. In addition, based on the definition of the term fundamental as “serving as a basis supporting existence or determining essential structure or function” (Merriam Dictionary) its use in the title is absolutely justified in our opinion.

The authors report, for the first time in Giardia, the complete knock out of the cwp1 gene in the two diploid nuclei of the parasite by using a sequential Cre/loxP-based approach and selection marker rescue strategies. The results of this experimental approach appear effective in allowing genomic editing. However, this sequential approach has not been analyzed in detail after each step, which makes the procedure difficult to follow. The authors should include an analysis of the expression of target gene after each round of the process to verify that the system works and, at the same time, to observe the effects of single allele disruption on the physiology of the parasite during cyst formation.

> *In figure S2, we provide solid genomic evidence for successful sequential gene ablation at each stage of the generation of line Δ CWP1.*

On the phenotype level we did in fact monitor deposition of the CWP1 protein on the surface of differentiated cells as the Reviewer suggests. Our observations were that elimination of the first two CWP1 alleles presented no obvious encystation abnormalities with respect to wildtype controls. This was not surprising and likely the result of a compensatory increase in the expression of the remaining two CWP genes that would thereby nullify any attempt at detecting a significant decrease in CWP1 transcript.

The first indication of an aberrant phenotype in fluorescence microscopy using a monoclonal antibody against CWP1 was detected in cysts depleted of 3 out of 4 alleles. In comparison to wildtype cysts, this line presents gross aberrations in CWP1 deposition and distribution, accompanied by compromised integrity of cyst walls. This partial CWP1 knock-out phenotype presents varying degrees of severity which we have amply documented in a new supplementary figure 1. Cysts developing from cells depleted of either 1 or 2 alleles show no obvious abnormalities in our assay. All data are now presented in supplementary figure 1 and are discussed in lines 79-83 and 438-41 of the revised manuscript. Due to the incorporation of these data, the order in which supplementary figures are cited has been amended in the revised manuscript.

The authors performed 3'-RACE and RT-PCR to probe the disappearance of the *cwp1* transcript (It is confusing. See below.).

> *We agree with the Reviewer and have amended any remaining discrepancies. Please see lines 82-83 and the legend to figure 1, part C. We have clarified that detection of CWP1 and control transcripts was done using RT-PCR, whereas CWP2 transcript detection was performed with 3'RACE.*

However, the methods designed to confirm this experimental evidence look insufficient. Why the authors do not use Western blotting to test for the total disappearance of CWP1 is unclear.

> *In this paper we demonstrate full ablation of CWP1 on three independent levels: a) genomic (figures 1B and S2), b) transcriptional (figure 1C) and c) translational (figures 1D, 3A, 4A and 5). Immunoblotting is inferior to direct protein detection by IFA in terms of sensitivity of detection and would therefore add no meaningful information to the already comprehensive data panel we present. All of this independent and direct evidence for a full ablation of CWP1 genes is borne out by the complete lack of a wall structure in Δ CWP1 pseudocysts.*

The use in immunofluorescence assays of a commercial antibody against CWP1 for Waterborn Inc., which has not been published by the company, is an important weakness of this work. The original antibody developed against a 65 kDa cyst wall product by Stibbs et al. in the early 1990's was used to clone the *cwp1* gene encoding a 26 kDa protein by Mowatt et al. in 1995, but the current antibody that the company sells is not longer the original 5C1 monoclonal antibody. For that reason, a full characterization of this reagent is needed before use in Western blotting and IF. On the other hand, fully characterized monoclonal antibodies against CWP1 and CWP2 have been generated and reported by several groups (Lujan, Faubert, Nash, etc.). Why the authors have not used those antibodies is unclear. TEM studies in which immuno EM with anti-CWPs antibodies look very feasible and necessary.

> We appreciate the reviewer's concern but the notion that there might be a weakness due to the choice of antibody with which we detect CWP1 is completely unfounded for the following reasons:

- 1) There is some confusion with respect to the Reviewer's reference to the antibody 5C1. This mAb DOES NOT recognize CWP1 but is in fact an antibody against a variant surface protein of trophozoites.
- 2) The antibody commercialized by Waterborne Inc. is mAb 5-3C, a widely used monoclonal antibody in the Giardia research field [1,2,3,4]. The two foremost experts on this topic, Dr. Theodore Nash (NIH) and Dr. Hugo Lujan (HHMI), have confirmed to us in writing that this is a highly specific anti-CWP1 mAb.
- 3) Following up on the Reviewer's concern about the mAb commercialized by Waterborne Inc., which was used in our study, we contacted the head of the company Dr Stibbs, for a detailed description of this reagent. We received confirmation that the anti-CWP1-TxRed mAb distributed by the company is:
 - a) a fluorescent conjugate of mAb 5-3C
 - b) binding to recognize CWP1 specifically
 - c) identical to the mAb used in [5] to clone CWP1.Correspondence with Dr. Stibbs is available upon request.
- 4) The data presented here are completely in line with mAb 5-3C being an anti-CWP1 antibody and provide unequivocal evidence for this antibody's specificity, given that :
 - a) mAb 5-3C was tested on cell samples lacking its target antigen i.e. cell line Δ CWP1, and, as expected, failed to give signal. This is evidence for specific antigen binding ;
 - b) mAb 5-3C did not detect endogenous or epitope-tagged CWP2 sequestered in the ER of Δ CWP1 cells, although the HA epitope-tagged reporter was clearly detected using an anti-epitope antibody. This indicates that mAb 5-3C does not crossreact with phylogenetically related proteins such as CWP2 and CWP3 and is therefore specific to CWP1.

In this scenario, knocking out CWP2, alone or jointly with CWP1, would provide better descriptions of the molecular basis of cyst wall formation in Giardia. So far, no antibody to CWP3 has been reported.

> Since there are two additional CWP family members it is quite obvious that there will be more possibilities for us and other groups in the field to analyze structure-function relationships based on the technical and conceptual milestones presented here. In the light of the complete block of cyst wall secretion shown here it would be interesting to know why the Reviewer thinks CWP2 would provide "better descriptions of the molecular basis of cyst wall formation in Giardia".

Again, although the strategy developed by the authors to knock out genes for the first time in this polyploid, binucleate parasite is of outstanding relevance in the field, the lack of appropriate controls require taking this technique with caution, at least for the moment.

> We thank the Reviewer for commenting about the outstanding relevance of our work. However, we strongly disagree with the Reviewer's reference to lack of appropriate controls without providing more specific information about this. Given all the data that was made available to document the ablation of four CWP1 alleles in the Δ CWP1 line and the phenotypic consequences we are at a loss how to respond to this comment. We have shown that a serial knockout is possible for CWP1. The procedure is in place and can be replicated for other targets by any suitably equipped lab.

Another important issue regarding this work is the fact that the authors claim that secretory granules that form de novo from the ER during trophozoite encystation are Golgi-like structures. Reports from several groups showed that the molecular chaperone BiP (Gottig et al. 2006; Touz et al. 2002) and the ER marker Yip1 (Stefanic et al., 2006; Wampfer et al., 2014) are present in these granules, that these granules do not contain the KDEL-receptor (Gottig et al. 2006; Elias et al. 2008), among other Golgi markers (Stefanic et al., 2006; Wampfer et al., 2014), indicating that the ESVs are generated from the ER by condensation of CWPs and specific sorting events (Gottig et al., 2006). This work is not the first evidence that ESV formation from the ER requires the presence of several molecules working in conjunction (Gottig et al., 2006; Touz et al. 2002, 2003), as stated in the discussion section. How can ER resident proteins be in the ESVs if they are Golgi-like organelles? How do the Golgi-like organelles lack more specific Golgi markers?

> Neogenesis of Golgi-like organelles and formation of secretory granules in Giardia has been an important part of the literature since our initial publication in Mol Biol Cell in 2003 [6] and was developed and peer reviewed in 9 additional papers published by our group in highly reputed journals such as "Traffic", "Journal of Cell Science", and "PLoS Pathogens" as well as the most recent textbook on Giardia biology. The Reviewer's comments and the questions refer exclusively to data presented in previous publications and have no bearing on the manuscript under review. All of the Reviewer's questions are addressed in a model presented in [7].

Gottig et al. (2006) have shown that a basic c-terminal extension only present in CWP2 is necessary for ESV biogenesis from the ER. Thus, knocking out CWP2 appears as the most relevant experiment to study ESV formation in Giardia and all other comments are speculative in essence.

> We chose to begin our analysis with CWP1 because it is a) the most abundant cyst wall protein and the only one deposited exclusively in the outer layer of the cyst wall, b) readily detectable using a highly specific mAb. The data presented here actually disagree with the Reviewer's evaluation that, in the absence of a CWP2 knock-out line, "all other comments are speculative in essence". In this work, we provide in fact robust experimental evidence for a clear set of morphological and molecular defects resulting from exclusive CWP1 ablation, including failure to export CWP2 from the ER.

In Konrad et al. (2010, PLoS Pathogens), we provided the experimental evidence for the hypothesis set forth in Gottig et al. (2006) by showing where and when during secretory transport post translational modification occurs in the CWP2 C-terminus and what some of its functions are.

Additionally, the detection of other cyst wall components in the so called “pseudocysts” devoid of CWP1 should be tested. For that, the available antibodies to untagged CWPs may provide unequivocal characterization of these structures and avoid speculations regarding Golgi characteristics.

> We used dually epitope-tagged CWP2 which was previously shown to be a suitable substrate for processing by the endogenous protease as a pre-requisite for ESV core condensation and maturation [7,8], suggesting that this reporter behaves exactly as its wildtype counterpart. By using using tEM to show absolutely no deposition of protein on the pseudocyst plasma membrane we provide unequivocal evidence that no secretion of CWPs has occurred in differentiated Δ CWP1 cells.

Previous reports from Hehl’s group have shown that different tagged versions of CWPs have distinct roles in cyst wall formation (Faso & Hehl, 2011). However, the use of untagged but detectable constructs seems essential since the tags can modify protein behavior, in particular during the dynamic process of Giardia encystation. Moreover, the authors claim that the “pseudocysts structures” are similar to that previously shown by Hehl’s group when disturbing ARF1 function. However, ARF1 is also present in the ER and in the lysosomal/endosomal compartment of Giardia (Lujan et al., 1995). Therefore, what is the link between ARF1, ESVs and CWPs? A clear discussion about the situation is needed.

> The essential role of the small GTPase ARF1 for secretion of CWPs to the surface of differentiating Giardia was described in detail in our paper Stefanic et al., J Cell Sci. 2009 [9]. The wall-less cysts produced by functional ablation of ARF1 are not the same as the pseudocysts described here but they share three hallmark features: 1) four nuclei, 2) disassembly of the adhesive organelle and flagella, 3) lack of a wall structure and concomitant loss of osmotic resistance. However, the pseudocysts shown here differ fundamentally as neogenesis of the entire regulated secretory pathway necessary for formation of a cyst wall, including ESV organelles, is missing. Thus, without ESVs, ARF1 does not come into play at all.

The Reviewer’s question refers to work published previously.

The authors also stated that, citing recent studies of Hehl’s group (Konrad et al., 2010), correct condensation of CWPs is necessary for segregation into the ESVs. But that was proposed earlier by Gottig et al. in 2006.

> Ours was the first group to detect and characterize condensed cores in ESV organelles [10], but we thank the reviewer for pointing out a missing reference. We have amended our text at line 46 and included the suggested reference in the revised version of the manuscript.

Moreover, given that previous reports showed important difference in the interpretation of the results between these two groups, the authors of this manuscript should comments on the discrepancies and perform additional experiments to provide new insight into this scientific disagreement.

> Different opinions and interpretations are ubiquitous in every field. Because this is not a review article we are not able to discuss issues which pertain exclusively to previously published literature and have no bearing to the manuscript under review.

The authors also claim that CWP1 is released to the culture medium and that binds the carbohydrate polymer of the cyst wall, but what about CWP2 (or even CWP3). The minor differences between the LLR domains of these proteins do not guarantee that the only one tested is the only one working (Chatterjee et al.; 2010). No comparative study between the LRR of CWPs has been reported.

> Actually Chatterjee et al. [11] show that native CWP1 binds the unique cyst wall glycan better than the other CWPs. Thus, we used native CWP1 as a readily available and detectable reporter for experimental testing of the question whether cyst wall glycan could be on the surface of Δ CWP1 pseudocysts. CWP2 or 3 might also bind but including them would not provide more information regarding the specific question we wanted to address. In addition, these CWPs are not as readily detectable and therefore not as convenient as reporters. Following the Reviewer's suggestion we have incorporated a reference in the text to recent work [12] demonstrating shedding of native CWP1 in the culture medium and binding to cyst walls.

Another important consideration regarding this work is the use of episomal vectors to rescue the knock out phenotype. In most protozoa, expression of a given protein is not universal with episomal vectors. Thus, how do the authors observe a complete rescue of the pseudocyst phenotype?

*> We are unclear as to what the reviewer intends with the term « universal ». Expression from episomally-maintained constructs in *Giardia lamblia* was chosen over integration of a single CWP1 gene because gene dosage is higher due to the presence of multiple plasmids in a nucleus. If expression of the complementing gene is too low, the phenotype is not complemented fully. However, negative feedback regulation ensures that the upper limit of CWP expression is tightly controlled. The necessary cis-acting elements are in fact contained in the episomal vector used here and allow generating sufficiently high levels of CWP1 expression for full rescue of the mutant phenotype in induced cells. The experimental readouts for the rescue are straightforward and include formation of ESVs (figure 4) and the appearance of a cyst wall (figure 4), both detected by the anti-CWP mAb in fluorescence microscopy.*

The authors also refer to a work of Poxleitner et al.; 2008, in which by using either linearized DNA or episomal vectors only one nucleus of the parasite is targeted. However, they do not need to modify their protocol to have successful results. On one hand, they stated that the efficiency of their strategy needs further improvements and, on the other hand, they claim complete success. It needs to be clarified.

>The observation made in the Poxleitner paper could be interpreted as refractoriness of one nucleus to transfection, a potential bottleneck for knockout of alleles three and four. However, in this study we performed sequential locus ablation combined with very powerful antibiotic resistance markers to select for transfections of the apparently more refractory nucleus. It turns out that this as-yet-uncharacterized refractoriness can be overcome using this approach.

In our manuscript, we write (lines 292-4): „Although our data points to its effectiveness, the sequential knock-out strategy could be further improved by addressing two main bottlenecks, concerning a) loss of Cre-encoding plasmid and b) efficiency of Cre/loxP mediated excision.”

Thus, we can in fact claim “complete success” but nevertheless suggest at least 2 aspects of our method that could be improved.

From the 47 references of the manuscript, almost a half of them are from the Hehl’s group while results and hypothesis reported by other groups are completely neglected. The authors should acknowledge the work of others and seriously discuss their scientific differences.

> We checked our reference list and found redundancies bringing the overall number of references down to 44 (these have been corrected in the revised version of the manuscript). Contrary to the Reviewer’s concern, only 8 of those were directly reported by the Hehl group. In our opinion, and given that we have worked and published on this topic for 16 years, this is a completely appropriate proportion. For the sake of scientific transparency, we have striven to acknowledge relevant work coming from all groups and have included references to reports that may not be in full agreement with our data.

Figure 1C: It is confusing since in the text the authors indicated that they perform 3’-RACE but in the figure legend they show an RT-PCR analysis. The primer 1040 is for RACE, as indicated in Table 1 as k-adaptor.

> We thank the reviewer for pointing this out and, as previously discussed, we have corrected these inconsistencies. We have clarified that detection of CWP1 and control transcripts was done using standard RT-PCR, whereas CWP2 transcript detection was performed with a more sensitive variant of this method, i.e. 3’RACE.

This can be easily clarified but the problem is that the CWP1 transcript is present in NON-ENCYSTING and PRE-ENCYSTING trophozoites, when they stated in the Abstract “CWP1 is expressed only in encysting cells”; in line 40 “the genes coding for CWPs are completely silenced in trophozoite”; and in line 277 “By targeting CWP1, we took advantage of the STRCT stage-specific regulation of CWP1 expression”. These contradictions are objectionable.

> Any log-phase population of trophozoites cultured in vitro contains a few percent of cells which encyst spontaneously. This was mentioned by several groups independently [10,13,14,15,16] and was analyzed in more detail in [17]. Thus, any sampling of RNA from a trophozoite population will reveal a small background signal for CWP transcripts. It is important to note that this signal does not indicate leaky promoters but a small proportion of trophozoites which undergo full encystation spontaneously. Thus, it is correct to say that the CWP1 locus is silenced in non-encysting trophozoites.

In Fig. S2, the results of amplification of primer pairs 2414/2415 and 2416/2417 are missing. Please clarify why.

> The PCR settings chosen for amplification using primer couple 2414 and 2415 were such that we did not expect any product greater than 600bp i.e. our target product, to be synthesised. Nevertheless, the reaction still allowed for amplification of a 3kb product derived from the second disrupted locus and is present only in samples KO2, KO3 and KO4, as predicted. The corresponding products derived from WB (ca. 8kb), KO3 (ca. 7kb) and KO4 (ca. 8kb) samples were too large to be amplified in this reaction, explaining why they are not present. The same reasoning can be applied to PCR reactions performed using primer couple 2416 and 2417, where our target product was again ca. 600bp large. Corresponding products

derived from the WB (ca. 6kb), KO3 (ca. 4kb) and KO4 (ca. 6kb) samples are too large to be amplified in these conditions.

Finally, the authors stated that the CRISPR/Cas9 system does not work in *Giardia* based on their failure to edit the parasite genome using that valuable tool. Many sentences in the abstract, results and discussion sections (in addition to several references) are based on their negative results (not shown). It is unclear why the authors reached their conclusion based on a technique that is not described at all in the manuscript. It is not necessary to elaborate on a technique that did not work in the authors' hands to make their editing approach more important. If their system work as they claim, why discourage other groups to attempt to make CRISPR/Cas9 to work in *Giardia*.

> Despite the many reports on the application of the CRISPR/Cas9 methodology to diverse species, there is currently no report on the feasibility of this technique in Giardia lamblia. This is telling in itself, given the hurdles associated to genome editing techniques in polyploid organisms and the impact that the CRISPR/Cas9 method could have on Giardia research. We and several other labs have invested considerable resources in developing CRISPR/Cas9 for Giardia, but with no success. Given the tremendous success of CRISPR/Cas9 in a tremendous range of organisms we need to explain why we use a comparatively laborious approach to knock out genes in Giardia. We certainly do not wish to discourage anyone from pursuing this ambitious goal. Nevertheless, the procedure presented in this manuscript is currently the only known method to achieve complete gene (and therefore protein) ablation in Giardia lamblia.

Reviewer #3 (Remarks to the Author):

The authors employed the Cre/loxP methodology, which they previously developed for use in *Giardia*, to knockout all 4 genes of CWP1 to explore the loss of this gene on the encystation, ESV formation and cyst wall development. Although the authors did not develop this methodology, they cleverly applied it for use in *Giardia*. This work took an enormous amount of work and diligence, and it is novel and of interest to cellular biologists working in secretion, Golgi genesis and function, encystation and persons working in *Giardia* and other protozoa that encyst.

> We thank the reviewer for their appreciation of our efforts and their general significance beyond the immediate field.

Because *Giardia* is so unique in the some aspects of its biology noted above, it is unclear how general the findings will apply to other organisms. One of the most interesting aspects of *Giardia* biology is the need for and functioning of two supposedly equally functioning nuclei. This system has the capability to knockdown both genes in only one nucleus so that the need and function of each nucleus, how they coordinate with each other and other interesting aspects of this bi-nucleated protozoa can be studied. One only needs to add epitope markers to tag each allele.

> We appreciate the Reviewer's insightful comments.

In *Giardia* an easy and sometimes very effective method to decrease and even eliminate transcription is expression of antisense transcripts to the gene of interest. Levels can be

undetectable. There was some discussion in an earlier paper from the same lab originally describing the Cre/LocP system in *Giardia* concerning the limitations of antisense methodology, but it may be very effective and save a lot of work and give the same answer. Perhaps the prior discussion can be repeated.

> Complete ablation of gene expression using knock down techniques was never reported so far in G. lamblia. Currently, the highest reported estimate for protein depletion was 90% in [18] as a result of long double-stranded RNA expression. However, this was the case for only one protein candidate, whereas depletion levels in other experiments are reported at anywhere between 22 and 70% [18,19,20,21]. To expand the context of the current paper we followed the Reviewer's suggestion and included the above information in the text, lines 272-4.

Obviously Cre/loxP has a number of advantages, but as seen in this work, it demanding and labor intensive. The biological effects of CWP1 knockout are interesting and well proven. However, the ins and outs of Golgi formation, cyst wall protein interactions, ESV formation and cyst wall protein formation will be hard sledding for the most readers. The paper is well presented and written. It appears to be technically correct and the *Giardia* methods clear and correct. In the testing of Δ cwp1 cysts for viability using impermeable dyes, clarify if the cysts tested with were exposed to water first or not.

> To avoid introducing any bias in the experiment, cysts were not exposed to water prior to the application of the viability stain. We have amended lines 411-13 to clarify this aspect of the method in « Freshly-harvested cysts were stained with acridine orange and ethidium bromide both at a final concentration of 100 μ g/ml in PBS [22], observed under a wide-field fluorescent microscope and scored for viability (>150 cysts/sample). Cysts were not exposed to water prior to staining ».

In the discussion of antibiotic resistance genes for *Giardia* (line274), there is mention of only two selectable antibiotic systems for *Giardia* but are there not three, blasticidin, puromycin and G418- all actually used in this work?

> We thank the reviewer for pointing this out. Neomycin and puromycin are considered to be the most reliable and effective of the three [23]. We clarified this aspect by amending line 274: «The availability of only two highly effective antibiotic resistance markers for selection of transgenes...».

The history of gene introduction and manipulation in *Giardia* starting on line 264 is not correct. The first published work was by Yee et al in June 1995. They used a plasmid containing luciferase with 5' and 3' *Giardia* GDH. Shortly thereafter in September of the same year Yu et al published a *Giardia* RNA virus construct; they added a neomycin selectable marker in 1996. There is no mention of antisense methodology by Touz et al first used in 2002. Singer et al and Sun simultaneously published on a stable expression vectors etc. but only Sun is referenced.

> We thank the Reviewer for pointing this out. We have re-organized this paragraph and included the missing reference in the revised version of the manuscript, lines 266-8. In this section, we refer exclusively to genetic engineering of G. lamblia on a genomic level; this is why we do not include the reference cited by the reviewer concerning the first use of antisense technology.

References

1. Einarsson E, Troell K, Hoepfner MP, Grabherr M, Ribacke U, et al. (2016) Coordinated Changes in Gene Expression Throughout Encystation of *Giardia intestinalis*. *Plos Neglected Tropical Diseases* 10: e0004571.
2. Merino MC, Zamponi N, Vranich CV, Touz MC, Ropolo AS (2014) Identification of *Giardia lamblia* DHHC proteins and the role of protein S-palmitoylation in the encystation process. *Plos Neglected Tropical Diseases* 8: e2997.
3. Nageshan RK, Roy N, Ranade S, Tatu U (2014) Trans-spliced heat shock protein 90 modulates encystation in *Giardia lamblia*. *Plos Neglected Tropical Diseases* 8: e2829.
4. Midlej V, Meinig I, de Souza W, Benchimol M (2013) A new set of carbohydrate-positive vesicles in encysting *Giardia lamblia*. *Protist* 164: 261-271.
5. Mowatt MR, Lujan HD, Cotten DB, Bowers B, Yee J, et al. (1995) Developmentally regulated expression of a *Giardia lamblia* cyst wall protein gene. *Molecular microbiology* 15: 955-963.
6. Marti M, Regos A, Li Y, Schraner EM, Wild P, et al. (2003) An ancestral secretory apparatus in the protozoan parasite *Giardia intestinalis*. *The Journal of biological chemistry* 278: 24837-24848.
7. Konrad C, Spycher C, Hehl AB (2010) Selective condensation drives partitioning and sequential secretion of cyst wall proteins in differentiating *Giardia lamblia*. *PLoS pathogens* 6: e1000835.
8. Gottig N, Elias EV, Quiroga R, Nores MJ, Solari AJ, et al. (2006) Active and passive mechanisms drive secretory granule biogenesis during differentiation of the intestinal parasite *Giardia lamblia*. *The Journal of biological chemistry* 281: 18156-18166.
9. Stefanic S, Morf L, Kulangara C, Regos A, Sonda S, et al. (2009) Neogenesis and maturation of transient Golgi-like cisternae in a simple eukaryote. *Journal of cell science* 122: 2846-2856.
10. Tolba ME, Kobayashi S, Imada M, Suzuki Y, Sugano S (2013) *Giardia lamblia* transcriptome analysis using TSS-Seq and RNA-Seq. *PloS one* 8: e76184.
11. Chatterjee A, Carpentieri A, Ratner DM, Bullitt E, Costello CE, et al. (2010) *Giardia* cyst wall protein 1 is a lectin that binds to curled fibrils of the GalNAc homopolymer. *PLoS pathogens* 6: e1001059.
12. Krtkova J, Thomas EB, Alas GCM, Schraner EM, Behjatnia HR, et al. (2016) Rac Regulates *Giardia lamblia* Encystation by Coordinating Cyst Wall Protein Trafficking and Secretion. *mBio* in press.
13. Hehl AB, Marti M, Kohler P (2000) Stage-specific expression and targeting of cyst wall protein-green fluorescent protein chimeras in *Giardia*. *Molecular biology of the cell* 11: 1789-1800.
14. Morf L, Spycher C, Rehrauer H, Fournier CA, Morrison HG, et al. (2010) The transcriptional response to encystation stimuli in *Giardia lamblia* is restricted to a small set of genes. *Eukaryotic cell* 9: 1566-1576.
15. Palm D, Weiland M, McArthur AG, Winiacka-Krusnell J, Cipriano MJ, et al. (2005) Developmental changes in the adhesive disk during *Giardia* differentiation. *Molecular and biochemical parasitology* 141: 199-207.
16. Franzen O, Jerlstrom-Hultqvist J, Einarsson E, Ankarklev J, Ferella M, et al. (2013) Transcriptome profiling of *Giardia intestinalis* using strand-specific RNA-seq. *PLoS computational biology* 9: e1003000.
17. Bernander R, Palm JE, Svard SG (2001) Genome ploidy in different stages of the *Giardia lamblia* life cycle. *Cellular microbiology* 3: 55-62.
18. Rivero MR, Kulakova L, Touz MC (2010) Long double-stranded RNA produces specific gene downregulation in *Giardia lamblia*. *The Journal of parasitology* 96: 815-819.

19. Mendez TL, De Chatterjee A, Duarte TT, Gazos-Lopes F, Robles-Martinez L, et al. (2013) Glucosylceramide transferase activity is critical for encystation and viable cyst production by an intestinal protozoan, *Giardia lamblia*. *The Journal of biological chemistry* 288: 16747-16760.
20. Huang YC, Su LH, Lee GA, Chiu PW, Cho CC, et al. (2008) Regulation of cyst wall protein promoters by Myb2 in *Giardia lamblia*. *The Journal of biological chemistry* 283: 31021-31029.
21. Garlapati S, Saraiya AA, Wang CC (2011) A La autoantigen homologue is required for the internal ribosome entry site mediated translation of giardiavirus. *PloS one* 6: e18263.
22. Ribble D, Goldstein NB, Norris DA, Shellman YG (2005) A simple technique for quantifying apoptosis in 96-well plates. *BMC biotechnology* 5: 12.
23. Gourguechon S, Cande WZ (2011) Rapid tagging and integration of genes in *Giardia intestinalis*. *Eukaryotic cell* 10: 142-145.

Reviewers' comments:

Reviewer #1 (Remarks to the Author):

The authors have made careful and thoughtful responses to the critiques of this reviewer and the others. While it would be interesting to "stretch" the experiments to include characterization of the GalNAc homopolymer in CWP1 knockouts and to test truncated versions of CWP1 in the complementation experiments, it is not necessary here. The "tour de knockout" experiments show that CWP1 is necessary for Giardia cyst wall formation in vitro, which is the major and important conclusion of the manuscript.

Reviewer #2 (Remarks to the Author):

I am highly disappointed by the authors regarding their responses to my concerns and request of changes in the manuscript, including the addition of new experiments. For example, Mowatt et al. (1995) and Lujan et al. (1996) demonstrated the lack of CWP1 mRNA in non-encysting trophozoites. Now the authors say the presence of such transcript in proliferating cells is common. I wonder if the authors are using the right culture media. If not, their entire work may be compromised. Additionally, if it is true that the anti-CWP1 antibody for Waterborne Inc. is Mab 5-3C, there should be no problem to perform the Western blotting experiments I requested.

Reviewers' comments:

Reviewer #1 (Remarks to the Author):

The authors have made careful and thoughtful responses to the critiques of this reviewer and the others. While it would be interesting to "stretch" the experiments to include characterization of the GalNAc homopolymer in CWP1 knockouts and to test truncated versions of CWP1 in the complementation experiments, it is not necessary here. The "tour de knockout" experiments show that CWP1 is necessary for Giardia cyst wall formation in vitro, which is the major and important conclusion of the manuscript.

> Thank you !

Reviewer #2 (Remarks to the Author):

Reviewer #2: I am highly disappointed by the authors regarding their responses to my concerns and request of changes in the manuscript, including the addition of new experiments.

> Below, we respond (once again) to the two remaining specific concerns raised by Reviewer #2.

Reviewer #2: For example, Mowatt et al. (1995) and Lujan et al. (1996) demonstrated the lack of CWP1 mRNA in non-encysting trophozoites. Now the authors say the presence of such transcript in proliferating cells is common. I wonder if the authors are using the right culture media. If not, their entire work may be compromised.

> *Detection of CWP1 transcripts in the two papers cited by the Reviewer was done in the mid-1990s using Northern blot-based methods which lack amplification and are therefore manyfold less sensitive than either RT-PCR or high throughput sequencing approaches.*

In our previous rebuttal, we already provided conclusive arguments to document observation of a low percentage of spontaneously encysting cells in in vitro cultures of log-phase populations of trophozoites made by several groups independently, and we attached the relevant references [1-8]. Analyses of mRNA levels by semi-quantitative RT-PCR as performed in this study will detect even minuscule amounts of CWP1 transcript in non-encysting and pre-encysting populations. More importantly, the phenomenon is also clearly documented in strand and non-strand –specific RNA-seq experiments, where a considerable amount of CWP1 sense transcript was detected in non-encysting trophozoite populations, as reported in [3, 8]. Below we have included a snapshot of these data retrieved with the giardiadb.org genome browser. The CWP1 gene model, GL50803_5638, is highlighted in yellow in the lower part of the image.

Furthermore, profiling CWP1 mRNA expression in different stages of the life cycle using a methodically different assay, i.e. serial amplification of gene expression (SAGE), clearly showed the same background expression of CWP1 sense transcripts during the trophozoite stage [7]; see red arrow in column graph below. This information is available on the CWP1 gene page:

(http://giardiadb.org/giardiadb/showRecord.do?name=GeneRecordClasses.GeneRecordClass&source_id=GL50803_5638&project_id=GiardiaDB) under the heading “Life Cycle SAGE Tags”. The red arrow indicates the small amount of CWP1 sense transcript detected in non-induced trophozoites.

Life Cycle SAGE Tags Hide

Taken together, there is ample evidence in the literature from numerous sources documenting that log-phase *Giardia* trophozoite cultures harbor the occasional spontaneously encysting cell. Sensitive methods will pick this up as a low CWP1 background signal at a population level.

With respect to cultivation of trophozoites, most if not all researchers in the *Giardia* field use a standardized complex culture medium (TYI-S-33) made according to a recipe published by Keister in 1983 [9]. We agree with the Reviewer that including this primary reference is important. Therefore, this reference has been cited at line 347 of the revised manuscript. Thus, if Reviewer#2 wonders whether we use the “right culture media” (unfortunately the Reviewer does not specify which one this might be) we would like to point out that cultivation of trophozoites in TYI-S-33 has been a standard in laboratories all over the world for more than 30 years.

Reviewer #2: Additionally, if it is true that the anti-CWP1 antibody for Waterborne Inc. is Mab 5-3C, there should be no problem to perform the Western blotting experiments I requested.

> After we invested considerable effort to refute this idea during the previous revision, it is strange and somewhat disconcerting that Reviewer#2 would still voice doubts about the specificity of mAb 5-3C (sold by Waterborne Inc.) against CWP1. To put this finally to rest we included confidential correspondence with Drs. Nash (NIH) and Stibbs (Waterborne Inc.) as an attachment to the cover letter.

The immunoblotting analysis using mAb 5-3C was performed as requested. We tested extracts prepared from trophozoites and encysting wild type and Δ CWP-1 cells. Data in Figures 1-3 already showed the absence of all four CWP1 alleles, as well as the CWP1 mRNA and exported cyst wall material in induced Δ CWP-1 cells. Not surprisingly, given the lack of a gene coding for CWP1 in Δ CWP-1, this experiment showed that the prominent 25 kDa band detected by mAb 5-3C in extracts of wild type cells is completely absent in the lane containing the Δ CWP-1 extract. Consistent with a low CWP1 background mRNA signal in non-induced populations, we detected a faint but specific signal in extracts from wild type non-encysting cells. This result is now included in Figure 1D and discussed in lines 88-93 and 119-21 of the revised manuscript. The methods and materials associated to this experiment are reported in lines 396-403 of the revised manuscript.

References:

1. Bernander, R., J.E. Palm, and S.G. Svard, *Genome ploidy in different stages of the Giardia lamblia life cycle. Cellular microbiology*, 2001. **3**(1): p. 55-62.
2. Chatterjee, A., et al., *Giardia Cyst Wall Protein 1 Is a Lectin That Binds to Curled Fibrils of the GalNAc Homopolymer. PLoS pathogens*, 2010. **6**(8).
3. Franzen, O., et al., *Transcriptome profiling of Giardia intestinalis using strand-specific RNA-seq. PLoS computational biology*, 2013. **9**(3): p. e1003000.
4. Hehl, A.B., M. Marti, and P. Kohler, *Stage-specific expression and targeting of cyst wall protein-green fluorescent protein chimeras in Giardia. Molecular biology of the cell*, 2000. **11**(5): p. 1789-1800.
5. Krtkova, J., et al., *Rac Regulates Giardia lamblia Encystation by Coordinating Cyst Wall Protein Trafficking and Secretion. mBio*, 2016. **in press**.
6. Morf, L., et al., *The transcriptional response to encystation stimuli in Giardia lamblia is restricted to a small set of genes. Eukaryotic cell*, 2010. **9**(10): p. 1566-76.
7. Palm, D., et al., *Developmental changes in the adhesive disk during Giardia differentiation. Molecular and biochemical parasitology*, 2005. **141**(2): p. 199-207.
8. Tolba, M.E., et al., *Giardia lamblia transcriptome analysis using TSS-Seq and RNA-Seq. PloS one*, 2013. **8**(10): p. e76184.
9. Keister, D.B., *Axenic culture of Giardia lamblia in TYI-S-33 medium supplemented with bile. Transactions of the Royal Society of Tropical Medicine and Hygiene*, 1983. **77**(4): p. 487-8.